# Using stakeholder insights to enhance engagement in PhD professional development

**Deepti Ramadoss**[1]☯*, **Amanda F. Bolgioni**[2]☯, **Rebekah L. Layton**[3], **Janet Alder**[4], **Natalie Lundsteen**[5], **C. Abigail Stayart**[6], **Jodi B. Yellin**[7], **Conrad L. Smart**[8], **Susi S. Varvayanis**[9]

**1** School of Medicine Office of Graduate Studies, University of Pittsburgh, Pittsburgh, PA, United States of America, **2** Department of Medical Sciences & Education, Boston University, Boston, MA, United States of America, **3** Office of Graduate Education, University of North Carolina, Chapel Hill, NC, United States of America, **4** Department of Neuroscience and Cell Biology and School of Graduate Studies, Rutgers University, Piscataway, NJ, United States of America, **5** Graduate School of Biomedical Sciences, UT Southwestern Medical Center, Dallas, TX, United States of America, **6** Office of Graduate and Postdoctoral Affairs, University of Chicago, Chicago, IL, United States of America, **7** Association of American Medical Colleges, Washington, D.C., United States of America, **8** Laboratory of Atomic and Solid State Physics, Cornell University, Ithaca, NY, United States of America, **9** Graduate School, Cornell University, Ithaca, NY, United States of America

☯ These authors contributed equally to this work.
* deepti.ramadoss@pitt.edu

**Data Availability Statement:** De-identified data is made available in accordance to IRB approved protocols. These file are available from the Open

## Abstract

There is increasing awareness of the need for pre- and post-doctoral professional development and career guidance, however many academic institutions are only beginning to build out these functional roles. As a graduate career educator, accessing vast silos and resources at a university and with industry-partners can be daunting, yet collaboration and network development are crucial to the success of any career and professional development office. To better inform and direct these efforts, forty-five stakeholders external and internal to academic institutions were identified and interviewed to gather perspectives on topics critical to career development offices. Using a stakeholder engagement visualization tool developed by the authors, strengths and weaknesses can be assessed. General themes from interviews with *internal* and *external stakeholders* are discussed to provide various stakeholder subgroup perspectives to help prepare for successful interactions. Benefits include increased engagement and opportunities to collaborate, and to build or expand graduate career development offices.

## Introduction

Institutions of higher education hold myriad potential connections between pre- and post-doctoral researchers, faculty and administrators, internal university offices, industry partners, professional societies, and funding organizations. Internal university partnerships are vital, ranging from pre- and post-doctoral researchers and university faculty, to externally-facing

Science Framework (OSF) database (DOI: 10.17605/OSF.IO/24SNV).

**Funding:** This work was supported by the Burroughs Wellcome Fund Career Guidance for Trainees [CGT025] (https://nam12.safelinks.protection.outlook.com/?url=https%3A%2F%2Fwww.bwfund.org%2F&data=04%7C01%7Cdeepti.ramadoss%40pitt.edu%7C70f047ae361445cf2e1b08d9c5d2fb33%7C9ef9f489e0a04eeb87cc3a526112fd0d%7C1%7C0%7C637758333513545052%7CUnknown%7CTWFpbGZsb3d8eyJWIjoiMC4wLjAwMDAiLCJQIjoiV2luMzIiLCJBTiI6Ik1haWwiLCJXVCI6Mn0%3D%7C2000&sdata=LCoaETVuHImoupB%2FEQCCp%2BTS%2BZaReZm9FBmcRHfkSTU%3D&reserved=0) (DR), National Institutes of Health [DP7OD020322] (https://nam12.safelinks.protection.outlook.com/?url=https%3A%2F%2Fwww.nih.gov%2F&data=04%7C01%7Cdeepti.ramadoss%40pitt.edu%7C70f047ae361445cf2e1b08d9c5d2fb33%7C9ef9f489e0a04eeb87cc3a526112fd0d%7C1%7C0%7C637758333513545052%7CUnknown%7CTWFpbGZsb3d8eyJWIjoiMC4wLjAwMDAiLCJQIjoiV2luMzIiLCJBTiI6Ik1haWwiLCJXVCI6Mn0%3D%7C2000&sdata=zijFvu9h2EVOjSfrvYuZqWGM47CaHA1rE7P%2BVc4Pah0%3D&reserved=0) and Burroughs Wellcome Fund, Career Guidance for Trainees [1018849] (https://nam12.safelinks.protection.outlook.com/?url=https%3A%2F%2Fwww.bwfund.org%2F&data=04%7C01%7Cdeepti.ramadoss%40pitt.edu%7C70f047ae361445cf2e1b08d9c5d2fb33%7C9ef9f489e0a04eeb87cc3a526112fd0d%7C1%7C0%7C637758333513545052%7CUnknown%7CTWFpbGZsb3d8eyJWIjoiMC4wLjAwMDAiLCJQIjoiV2luMzIiLCJBTiI6Ik1haWwiLCJXVCI6Mn0%3D%7C2000&sdata=LCoaETVuHImoupB%2FEQCCp%2BTS%2BZaReZm9FBmcRHfkSTU%3D&reserved=0) (AFB); National Institutes of Health [DP7OD020317] https://nam12.safelinks.protection.outlook.com/?url=https%3A%2F%2Fwww.nih.gov%2F&data=04%7C01%7Cdeepti.ramadoss%40pitt.edu%7C70f047ae361445cf2e1b08d9c5d2fb33%7C9ef9f489e0a04eeb87cc3a526112fd0d%7C1%7C0%7C637758333513545052%7CUnknown%7CTWFpbGZsb3d8eyJWIjoiMC4wLjAwMDAiLCJQIjoiV2luMzIiLCJBTiI6Ik1haWwiLCJXVCI6Mn0%3D%7C2000&sdata=zijFvu9h2EVOjSfrvYuZqWGM47CaHA1rE7P%2BVc4Pah0%3D&reserved=0) and National Institute of General Medical Sciences [1-R01GM140282-01] (https://nam12.safelinks.protection.outlook.com/?url=https%3A%2F%2Fwww.nigms.nih.gov%2F&data=04%7C01%7Cdeepti.ramadoss%40pitt.edu%7C70f047ae361445cf2e1b08d9c5d2fb33%7C9ef9f489

communications and alumni development offices. University career and professional development (CPD) programs also develop and rely on external partnerships, particularly with programming and resources designed for pre- and post-doctoral researchers. While CPD programs understand that these partnerships improve the pre- and post-doctoral training experience, provide pipelines for entry of pre- and post-doctoral researchers into the workforce, and lead to synergies and collaboration, the full value of these relationships may not be completely understood to internal and external partners. Our work explores the foundational value of internal and external intersections and how to best leverage them to prepare pre- and post-doctoral researchers for the workforce. The aim is to more efficiently and successfully coordinate relationships that meet all stakeholders' needs with a more thorough understanding of stakeholder objectives and the relative value of engagements.

This project is a spinoff of the National Institutes of Health Broadening Experiences in Scientific Training–NIH BEST [1] Consortium's Annual Meeting, in 2018. The Consortium (funded 2013–2019) was comprised of programs at 17 higher education institutions challenged by the NIH to develop innovative approaches to prepare pre- and post-doctoral researchers for a wide range of careers in the biomedical research enterprise. The Consortium's final Annual Conference (in 2018) explicitly invited collaborations, presentations, and conversations through joint programming with institutions beyond the Consortium–ranging from well-established pioneer pre- and post-doctoral professional development program institutions to newer and aspiring institutional programs interested in establishing professional development programs, as well as external private and non-profit collaborators. This research project emerged from the *massively audacious goal* identified by and adopted by the Blue Sky Visioning Mastermind Group that "all pre- and post-doctoral scholars have support and resources needed to explore and pursue all careers, and faculty and institutional leadership buy in to the importance of this mission."

The goals of this publication are to bring awareness within the higher education community about various stakeholders that commonly engage with graduate CPD, shed light on stakeholder perceptions of career development and engagement, broaden the composition of engaged collaborations, and provide engagement tools. This information can help institutions and individuals quickly self-assess and visualize strengths/opportunities with stakeholders for the purposes of CPD at their institution, and, over the long-term, build stronger relationships and partnerships.

## Defining stakeholders

The authors quickly realized that to attain their goal, a broad set of stakeholders would need to be consulted. Informed by the authors' experience in industry relations and engaging with higher education, stakeholders were identified, classified, prioritized, and consolidated into a rapid tool for stakeholder engagement (see Methods). **Table 1** displays internal and external stakeholder classification groups and subgroups: *internal stakeholders* include pre- and post-doctoral researchers, faculty/administrators, and external-facing staff; *external stakeholders* include non-profit and society partners and industry employers. Each stakeholder subgroup was approached with a specific set of questions to explore their perceptions of graduate career education and professional development, as well as their motivations for engaging with any of the other stakeholder groups, ultimately seeking advice for how to best engage them in CPD programming. Descriptions of desirable and required skills, as well as resources to share with pre- and post-doctoral researchers were sought.

**Internal stakeholders.** The group that receives the most frequent focus from CPD professionals includes pre- and post-doctoral researchers. Fostering stakeholder engagement from

e0a04eeb87cc3a526112fd0d%7C1%7C0%7C637758333513545052%7CUnknown%7CTWFpbGZsb3d8eyJWIjoiMC4wLjAwMDAiLCJQIjoiV2luMzIiLCJBTiI6Ik1haWwiLCJXVCI6Mn0%3D%7C2000&sdata=Ov7fbmaQgul6IdBM3NMR8ucsBwBkBCOHjkcqqbt6LLs%3D&reserved=0) (RLL); National Institutes of Health [1DP7OD020314] (https://nam12.safelinks.protection.outlook.com/?url=https%3A%2F%2Fwww.nih.gov%2F&data=04%7C01%7Cdeepti.ramadoss%40pitt.edu%7C70f047ae361445cf2e1b08d9c5d2fb33%7C9ef9f489e0a04eeb87cc3a526112fd0d%7C1%7C0%7C637758333513545052%7CUnknown%7CTWFpbGZsb3d8eyJWIjoiMC4wLjAwMDAiLCJQIjoiV2luMzIiLCJBTiI6Ik1haWwiLCJXVCI6Mn0%3D%7C2000&sdata=zijFvu9h2EVOjSfrvYuZqWGM47CaHA1rE7P%2BVc4Pah0%3D&reserved=0) (JA); University of Texas Southwestern Medical Center Graduate School of Biomedical Sciences (https://www.utsouthwestern.edu/education/graduate-school/) (NL); National Institutes of Health [DP7OD020316] (https://nam12.safelinks.protection.outlook.com/?url=https%3A%2F%2Fwww.nih.gov%2F&data=04%7C01%7Cdeepti.ramadoss%40pitt.edu%7C70f047ae361445cf2e1b08d9c5d2fb33%7C9ef9f489e0a04eeb87cc3a526112fd0d%7C1%7C0%7C637758333513545052%7CUnknown%7CTWFpbGZsb3d8eyJWIjoiMC4wLjAwMDAiLCJQIjoiV2luMzIiLCJBTiI6Ik1haWwiLCJXVCI6Mn0%3D%7C2000&sdata=zijFvu9h2EVOjSfrvYuZqWGM47CaHA1rE7P%2BVc4Pah0%3D&reserved=0) (CAS); National Institutes of Health [DP7OD18425] (https://nam12.safelinks.protection.outlook.com/?url=https%3A%2F%2Fwww.nih.gov%2F&data=04%7C01%7Cdeepti.ramadoss%40pitt.edu%7C70f047ae361445cf2e1b08d9c5d2fb33%7C9ef9f489e0a04eeb87cc3a526112fd0d%7C1%7C0%7C637758333513545052%7CUnknown%7CTWFpbGZsb3d8eyJWIjoiMC4wLjAwMDAiLCJQIjoiV2luMzIiLCJBTiI6Ik1haWwiLCJXVCI6Mn0%3D%7C2000&sdata=zijFvu9h2EVOjSfrvYuZqWGM47CaHA1rE7P%2BVc4Pah0%3D&reserved=0) (SSV) The funders had no role in study design, data collection and analysis, decision to publish, or preparation of the manuscript.

**Competing interests:** The authors have declared that no competing interests exist.

pre- and post-doctoral researchers is critical for providing effective programming [2]. CPD programs less frequently look beyond pre- and post-doctoral researchers to engage a full spectrum of university stakeholders within their own ecosystem.

Foremost among them is the faculty. Data from the BEST Consortium indicated that a large majority of faculty are supportive of career training for various careers and have recognized that pre- and post-doctoral researchers participating in CPD activities were happier and making timely progress toward degree completion–a fact that can be used to recruit additional *internal stakeholder* engagement from faculty [3–5]. Faculty don't always believe they have the knowledge or resources to assist pre- and post-doctoral researchers whose career interests lie outside academia, although they largely support their career pursuits [5–8]. Promoting transparency, encouraging the normalized need for career support, and recommending conversations to initiate CPD coaching will help bridge some knowledge and awareness gaps.

In addition to partnerships with internal communications, environmental health and safety, or research support offices for experiential opportunities, identifying internal collaborators with external-facing roles has the potential to reduce the need to develop *de novo* program components, to seed ideas that leverage each other's networks and knowledge domains, and to overcome common roadblocks [9]. Cost-sharing on resources and events are an added benefit for engaging with internal partners (including student leaders) and opens doors for pre- and post-doctoral researchers to engage in on-campus job shadowing and internship opportunities for experiential learning [2, 9, 10]. Collaborations with alumni relations and development offices can dramatically expand the network of potential speakers and mentors [11]; it can also lead to coordinated fundraising campaigns for CPD initiatives. Moreover, establishing relations with other internal partners with external-facing roles in industry or federal relations, technology transfer, licensing, or research commercialization can amplify the skill sets of pre- and post-doctoral researchers seeking experiential learning opportunities in real-world settings [10].

**External stakeholders.** Efforts of career development professionals must simultaneously be internally and externally focused to fully understand the skills current pre- and post-doctoral researchers need to execute an informed transition into careers of their choice [12]. In addition, external focus identifies potential employers to build pipelines for these researchers, attract funding sources, access training opportunities to support their CPD, and increases visibility and accessibility of the resources offered by CPD programs. *External stakeholders* include partners and employers as seen in **Table 1**. The categorizations in **Table 1** are based on how CPD practitioners primarily interact with each group but can overlap with other categories, as many partners are also employers.

Stakeholders in external groups such as industry, non-profits and government agencies, including professional societies and associations, have long partnered with academia in disseminating research and technical training. They increasingly offer skill-building and career development opportunities via conferences and webinars to assist current pre- and post-doctoral researchers in the career selection process. Additionally, societies acknowledge multiple career options and wield positional influence to support culture change within academia.

A primary function of CPD programs is to strengthen the future workforce by preparing pre- and post-doctoral researchers for interaction with *external stakeholders*, ultimately, future employers. Therefore, the foci of these outward-facing efforts should be strategic to broaden networks and facilitate connections. Engaging external employer stakeholders in networking events, site visits, job shadowing, internships and panel discussions makes it possible for pre- and post-doctoral researchers to explore and test-drive various PhD careers [9, 10, 13]. This study provides what we hope are useful tools and insights to address all types of stakeholder engagement.

**Table 1. Stakeholder classification and examples.**

| Stakeholder Group | Sub-Group Interviewed | | N = 45 | Examples |
|---|---|---|---|---|
| Internal Stakeholders | 1) | Pre- and post-doctoral Researchers | 9 | Pre-doctoral students, Post-doctoral researchers |
| | 2) | Faculty/Admin | 8 | Assistant, Associate, Full Professors, Chairs of Departments, Directors of Research Centers |
| | 3) | External-Facing Staff | 12 | Staff administrators with roles in career services, industry / government relations, technology transfer / licensing, communications |
| External Stakeholders | 4) | Non-profit/society Partners | 8 | Trade organizations, professional societies, non-profits, business associations |
| | 5) | Industry Employers | 8 | Small and large companies, intellectual property firms, consultancies, accelerators |

## Methods

### Stakeholder identification and engagement

**Study design.** This study is intended as an exploration of a field, and an opportunity to observe emerging phenomena. The research team identified both *internal* and *external stakeholder* groups relevant to graduate CPD programs in order to further identify values that each stakeholder places upon bidirectional interactions to advance CPD for pre- and post-doctoral researchers. Common themes were identified through open-ended questions and unanticipated value propositions to develop potential approaches for improving interactions and methods of engagement between the university and potential partner organizations.

**Data collection.** Interviews offer a rich and robust capture of perspectives [14], providing in depth, open-ended responses and opportunity for dialogue between researcher and participant. Single-interaction, semi-structured interviews, conducted either in-person (before COVID-19), by phone, or online via Zoom, were used with standardized questions and optional probing follow-up questions as needed (see **S1 File** for complete interview question list). Once core questions were established, subsets of parallel but slightly amended questions relevant to each stakeholder group were developed. Due to privacy concerns, neither recording nor transcriptions were requested/approved for human subjects. Instead, interviewers took digital or manual live notes. In all interviews, verbal consent was sought from participants.

**Recruiting/access to and selection of participants.** Selection criteria for interview subjects were designed to be representative of identified stakeholder groups, with an initial goal of including 5–6 individual interview participants per stakeholder subgroup. *External stakeholders* initially included industry and non-profit/societies grouped together, but after the first few interviews, the research team discussed the difference in themes that arose from these interviews and arrived at the consensus to classify them as two distinct groups. Therefore, additional participants were recruited to ensure there were 5–6 participants for each of these subgroups.

Potential participants were identified by several means: by each interviewer independently, based on individuals known to the interviewers; referrals within and across the authors' networks; or vetted Google searches. The invitation selection process considered a broad variety of types of organizations or groups based on stakeholder classification (see **Table 1**) and identified a sampling of individuals within a particular subgroup that included variety in perceived support for the premise, academic disciplinary background, education level, as well as across social identity categories including gender, race/ethnicity, and international status. Other factors considered included leadership/experience level, tenure in respective organizations, and any experience or interest in working at the interface of professional/career development for varying purposes. This background was not known for all and some were purposefully naïve or underexposed to CPD initiatives in higher education. A review of prospective participants'

general characteristics served to uncover similarities or duplications. Efforts were made to ensure that a variety of types of organizations were represented in each subgroup.

**Conducting interviews.** Once selected, prospective participants were invited by email to participate in a short 15-20-minute interview. The template invitation (see **S2 File**) included a brief script of the study's purpose and description, plus an overview of the questions to be asked. Interviewer and participant found a mutually convenient time and format (in-person, phone, or video conference call).

After identifying stakeholder groups and subgroups (see **Table 1** and Discussion for the importance of flexibility and refinement), four interviewers conducted a total of 45 stakeholder interviews (of 55 invitations). Themes were collected separately for the groups that used the same sets of questions to differentiate responses (i.e. pre- and post-doctoral researchers versus faculty and administrators, external partners versus employers). Following the first round of interviews, one group had a higher sample size than the others, therefore, to keep group numbers roughly equal, target recruitment goals were updated to a minimum of eight interviews per subgroup (see **S1 Table**).

**Data analysis and interpretation/validity.** A multi-stage process was used for data analysis and interpretation, including sorting of sensitizing concepts [15], and analysis and reduction of data through application of grounded theory [16, 17], leading to the identification of emergent themes, hierarchical grouping, and concept categorization. One member of the research team who did not conduct any interviews was designated as coder. The coder and each individual interviewer reviewed and 'binned' potential initial themes emerging from keywords and phrases, and separated text into categories using paraphrased concepts or the original words from each participant into each row of a spreadsheet with category column headings. This ongoing collaborative synthesis of data and collection of emergent themes contributed to the iterative data reduction and display process, including a process of contrast/comparison, and noting patterns and themes [18].

At the conclusion of coding of each interview, coder and interviewer reviewed the initial data-sorting and 'binning' to ensure themes were appropriate and consistent with participant intent. With each subsequent interview within a stakeholder group (*internal*, *external*) and subgroup, themes were refined; new 'bins' were created if the participant's comments did not fit into an existing bin. If a response fit into two themes, then they were placed in both bins and coded as repeated. A second text review after all interviews were complete was conducted by the coder, with all authors working collaboratively in a process to establish inter-rater reliability. The team appraised the interviews in the larger context to make sure the original interview notes and emergent themes were not in conflict, still represented participant viewpoints, and to catch any themes missed in the first review. A final complete review by all authors prior to summation repeated this process through robust group discussion and collaborative decision-making [19]. All final themes and comments were reviewed and consolidated to assure researcher agreement on the accuracy of the themes and statements were selected to represent each theme.

Unique themes found in interview text were highlighted and reported as representative themes that arose when multiple instances of each theme occurred within a stakeholder subgroup.

**Subjectivity/ethical issues/limitations.** All interviews were conducted by researchers who are professionals in higher education (e.g., program directors, associate directors, assistant deans) strongly invested in CPD for pre- and post-doctoral researchers (within offices of graduate education, postdoctoral affairs, CPD programs, evaluation). All interviewers are professionally full-time employed women in the US, and the study team included US and international interviewers, both people of color, Asian, and White.

The possibility of selectivity bias exists, in opinions or stories shared by participants based on their roles, and in the selection of participants within individuals' networks tending toward supporters of graduate CPD. To attain a balanced view of various stakeholder subgroup perspectives as well as in recruiting participants, three methods were used to avoid compounding selectivity bias: 1) a semi-structured interview style with pre-selected questions (see Recruiting). 2) explicit recruitment of "nay-sayers" as well as "supporters" of university/organizational partnerships and graduate career training. 3) Online searches to identify further participants beyond known networks (e.g. Google), as well as searches and requests to members of known networks (e.g. LinkedIn) to suggest individuals the authors had no previous connection to, who might not be interested in or knowledgeable specifically about graduate professional development, but were in positions related to industry-university engagement activities. Each interviewer conducted interviews with individuals they knew and those they had never met, from offices or organizations they were familiar with and those with which they had no prior knowledge. Specific inclusion of the question, "Do you follow the national conversation about career development and outcomes of PhD-trained scientists?" helped determine their level of awareness of the subject matter. Nonetheless, the authors recognize the need to interpret findings with caution and that they will not represent all possible viewpoints. The results should be viewed as pilot data to inform additional research.

All participants provided oral informed consent. Participant names are anonymized using pseudonyms and all data is de-identified prior to sharing in accordance with IRB approved protocols (Rutgers–FWA00003913 Study ID Pro2020222400; PittPRO STUDY19110306-I4; UNC IRB# - 19–3054; Boston University H-40210). Note that the pseudonyms were randomly assigned and their perceived ethnicity or gender is not intended to represent the individual participant. Any correlation with a theme and gender, race or ethnicity is purely coincidental. Demographics were gathered at the last step to fill in post-analysis to prevent unconscious bias or revealing of identity or demographics of any participant. In detailed results in **S3 File**, only pseudonyms are used, using the code in **S2 Table**.

## Stakeholder engagement tool

The authors observed a gap between the perceived awareness of the variety of stakeholders and the ability to assess and capitalize on strengths of potential existing partnerships with *internal* and *external stakeholders*. To help rapidly assess the two, a tool was created. During the development of the stakeholder engagement tool, categories were refined based on discussion among the authors, their combined experiences working with various stakeholders, as well as a two-way influence of the interview process (the tool influencing the interviews, and the stakeholder perspectives from the interviews influencing refining the tool).

The authors first debuted the tool publicly during interactive workshops at sequential meetings of the 2020 international conference hosted by the Graduate Career Consortium. Input from conference attendees participating in that workshop helped question previous assumptions and helped to better describe how users would customize the tool.

**Use of the stakeholder engagement tool.** STEP 1: With your unit or collaborators, create a list of stakeholders—start with the suggested categories (see **Table 2** below) and customize for your institution & geographic area by entering your own labels to tailor to your institution. Try to be specific (i.e. don't say 'alumni office' but rather, 'alumni association of Boston'). You can further break these down into lists of names and contact info. Note that engaging with naysayers/agnostics can be as valuable as engaging with your supporters to identify blind spots and valuable alternative perspectives.

**Table 2. Suggested stakeholders.**

| Internal stakeholders/ users | faculty supporters, faculty agnostics/nay-sayers, admin staff, graduate students, postdocs |
|---|---|
| Internal partners | career services, industry relations, licensing office, communications office, alumni office |
| External partners | economic development, chamber of commerce, trade organizations, professional societies, funders |
| Employers | foundations/non-profits, local companies, accelerators/incubators, intellectual property firms, biopharma |

STEP 2: Based on your institutional knowledge, take one or two minutes with your team to quickly rank your perceived level of engagement with the stakeholder groups in the first column. The authors encourage users to define their own Likert scale to calibrate the level of engagement or use the suggested Likert scale in **Table 3**.

For analysis and interpretation of the stakeholder engagement tool, see the Results section.

## Results

A total of 45 individuals were interviewed by four interviewers (**S1 Table**). Consenting participants consisted of both men (42.2%) and women (57.8%). Participant demographics included US (71.1%) and international (28.9%); African American (15.6%), Asian (17.8%), White (62.2%), and Hispanic (4.4%). Participants were geographically diverse across the US, with interviewers accessing their own networks primarily across the Northeast, Southeast, and Midwest United States, but also extending geographic representation via national organization contacts and referrals.

Stakeholder organizations included public, private, and public-private hybrid institutions of higher education (some with medical schools); large and small companies in or serving the biotech, medtech or pharma industries; as well as foundations, non-profit organizations or business associations serving STEM fields. Major themes arising from each set of stakeholders are presented, with details available in **S3 File**.

### Stakeholder 1 –Internal pre-doctoral students and post-doctoral researchers

Interviews (**S1 File**) probed attitudes towards devoting time to CPD. Responses from 'frequent users', 'occasional users', or 'non-users' of CPD programming were separated. Responses broadly had two flavors: support for CPD programs and their perceived benefits, or opportunities to improve (**Fig 1**).

**Support for and perceived benefits of CPD programs identified by pre- and post-doctoral researchers.** Pre- and post-doctoral researchers find CPD affects productivity in a positive way and allows for CPD programming users to get a broad overview of resources (12 mentions or 12 M). Community-building advantages of professional development activities,

**Table 3. Suggested likert scale.**

| 0 | No interaction, little/no idea whom to contact |
|---|---|
| 1 | Have a short list of contacts, need to reach out, little/no personal connection |
| 2 | Reached out to one or more stakeholders, a few productive discussions |
| 3 | Know of and work with groups/people to engage, occasional activities together |
| 4 | Our offices/institutions/companies engage on a semi-regular basis |
| 5 | Well established, regular two-way engagement |

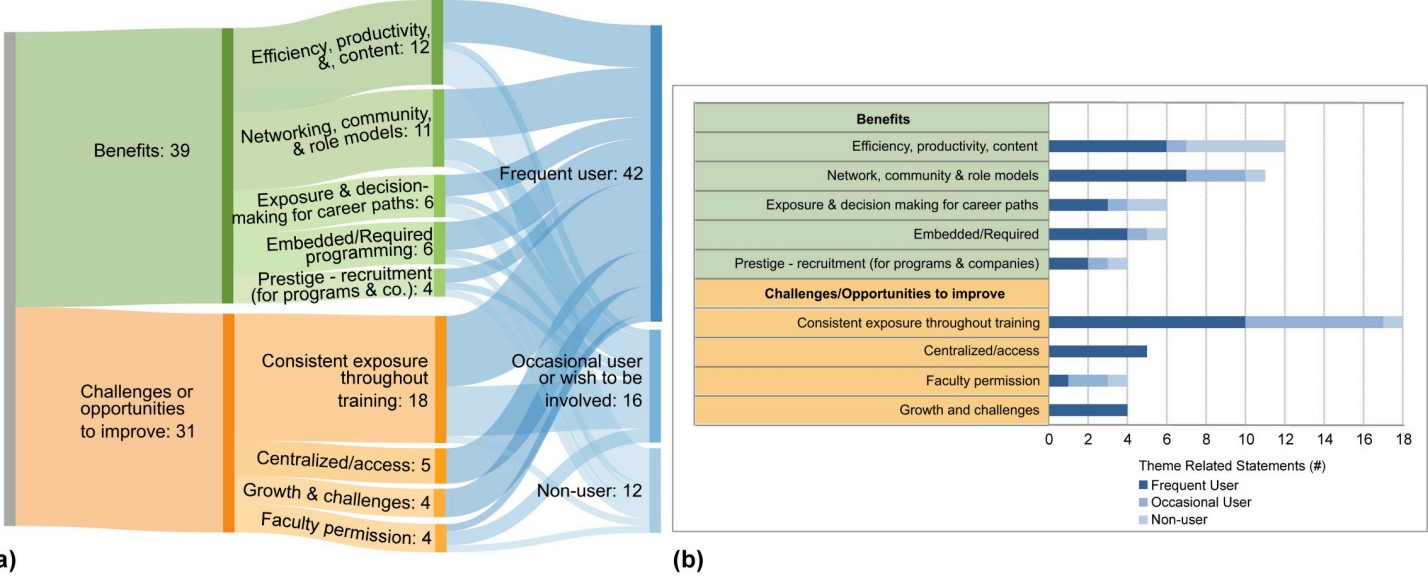

**Fig 1. Themes from internal pre- and post-doctoral researcher stakeholders.** (a) Sankey diagram and (b) stacked bar graph representing the same data of the number of mentions for each theme representing benefits and challenges/opportunities to improve mentioned by frequent users, occasional users and non-users.

e.g. bridging researchers across labs was discussed (11 M), highlighting confidence-building, cultural, and gender-based inclusivity that these programs can provide via intentional conversations and making role models prominent. This aligns with findings of a sense of community among those attending CPD training, especially in cohort or mandatory participation models [2]. The benefit of networking activities with hiring managers and recruiters outside academia was underscored. Advantages such as exposure to different career paths, allowing for informed decision-making (6 M), embedded/required CPD training providing consistent support and messaging (6 M), and prestige for recruitment (4 M) were raised.

**Challenges or opportunities to improve identified by pre- and post-doctoral researchers.** Pre- and post-doctoral researchers communicated their desire for consistent exposure to CPD activities throughout training (18 M). Centralized programming to equalize access to resources and institutionalize the concept of career development to facilitate faculty acceptance was viewed as necessary (5 M). Challenges around personal growth (4 M), and the express need for faculty permission (4 M) were identified.

For details, including representative comments for individual themes, see **S3 File**.

## Stakeholder 2–Internal faculty and administrators

Participants answered the same questions as the pre- and post-doctoral researchers subgroup above (**S1 File**) focused on probing attitudes towards devoting time to CPD, and opinions of existing or hypothetical CPD opportunities. Themes that arose naturally divided participant responses into categories that were later identified as 'enthusiastic supporter', 'cautious supporter', and 'non-supporter' responses (**Fig 2**).

**Perceived 'benefits' of CPD identified by faculty/administrators.** The benefits of CPD activities identified by faculty and administrators acknowledged evolving training requirements and climate (12 M), with comments ranging from requirements for training grant applications to how participation in these activities can improve mental health. Faculty and administrators acknowledged institutional peer pressure nationally among top tier universities

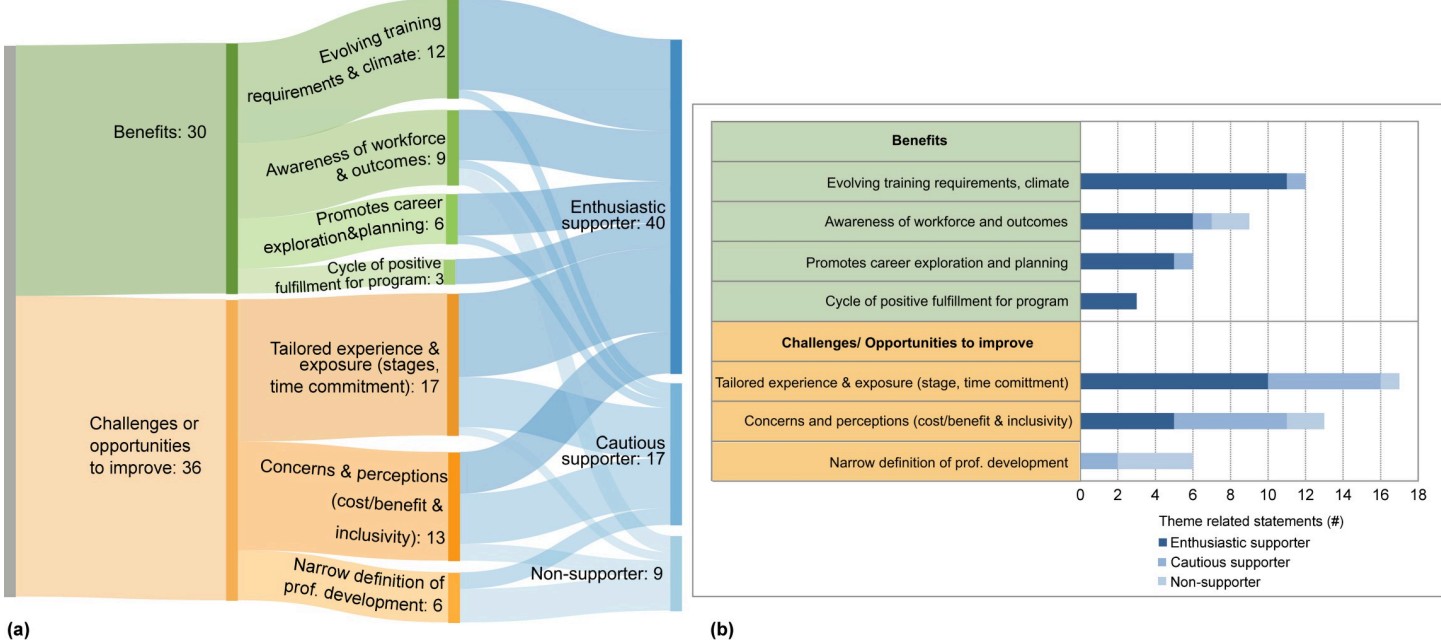

**Fig 2. Themes from internal faculty/administrator stakeholders.** (a) Sankey diagram and (b) stacked bar graph representing the same data of the number of mentions for each theme representing benefits and challenges/opportunities to improve mentioned by enthusiastic supporters, cautious supporters and non-supporters.

to provide these opportunities, and that as educators, they believe there is an obligation to provide these experiences.

Enthusiastic supporters acknowledged the limited availability of faculty positions, and the value of transparency of workforce outcomes of alumni (9 M). Less supportive respondents believe that professional development is not related to graduate education, that it de-emphasizes academia as a career path, and that individuals who leave academia cannot return. Enthusiastic and cautious supporters noted valuable effects of CPD programming, such as promoting career planning and gaining professional skill sets (6 M), and improved reputation of graduate programs (3 M).

**Challenges or opportunities to improve identified by faculty/administrators.** Multiple comments arose regarding early exposure to options, and the need for experiences tailored to each individual's stage and priorities (17 M). Some faculty raised concerns and perceptions (13 M) such as the belief that CPD could be a distraction, and lengthen time-to-degree completion. Narrow definitions of professional development revolving only around academic skills (6 M) are among challenges observed by interviewees.

For details, including representative comments for individual themes, see **S3 File**.

## Stakeholder 3: External-facing staff (industry relations, tech transfer, communications, alumni relations and development)

Interviews with external-facing staff were guided by a different set of questions (**S1 File**). The goals of these questions were to identify with whom external-facing offices at institutions typically interacted, and around what topic(s) the majority of their interactions centered. While these offices are open to all institution affiliates, it is not known if the external contacts have an interest in STEM pre- and post-doctoral researchers.

Respondents in external-facing offices interact with small and large businesses and foundations (8 individuals), investors (4 individuals), alumni (3 individuals), inventors (2

individuals), and innovation centers (2 individuals). These respondents often wear several hats, and interact with many other affiliates and identities including experts, educators, presenters, business development and intellectual property professionals, an institution's business school (e.g. on consulting projects), grateful patients, hospital systems, government, and professional organizations. It is of note that the themes below do not include a federal relations point of view (**Fig 3**).

**Reasons to engage with industry identified by external-facing offices.** Bringing the voice of the market to academic research is a strong motivation for interactions between industry and academia (16 M). Building partnerships between the two (11 M) has several benefits such as fundraising, increasing awareness of internship programs, and co-creation of curriculum content, and furthermore, provides long-term relationships and resources. External-facing staff are motivated by the possibility of fundraising or financial support (10 M) for the institution, as well as the perceived benefit to pre- and post-doctoral researchers (4 M).

**Reasons external stakeholders engage with academia identified by external-facing staff.** A recurring theme that emerged was the perceived interest of *external stakeholders* in early access to emerging technologies and innovations (18 M). *External stakeholders* are believed to be keen to assist in developing an entrepreneurial mindset (11 M), such as instilling skills to better serve both researchers and the workforce more broadly, with an external-facing staff member referring to a joint research brief on entrepreneurial mindset, Ernst & Young & Network for Teaching Entrepreneurship [20], and to reports published by the EU Commission

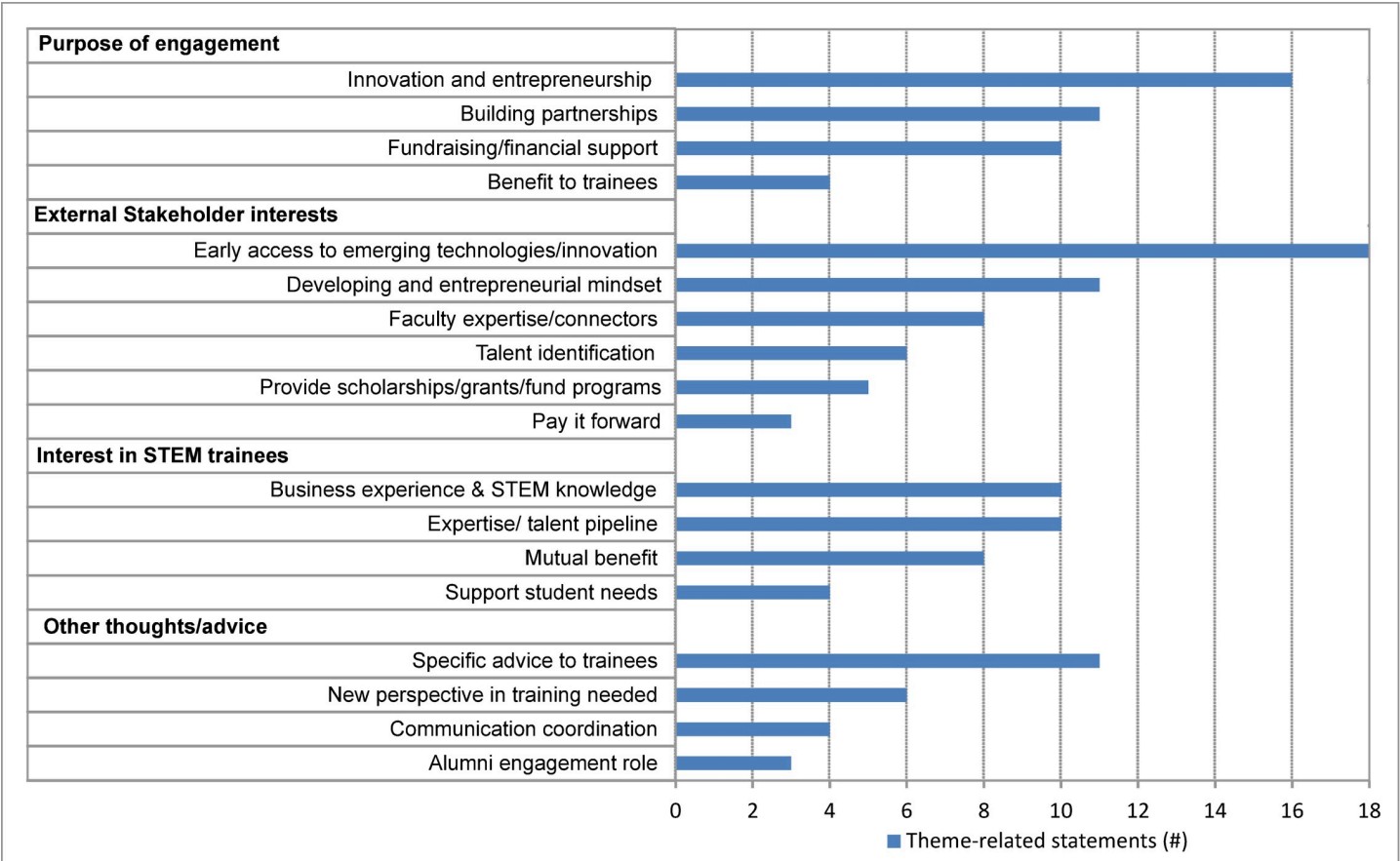

**Fig 3. Themes from external-facing staff.** Bar graph representing the number of mentions of each theme.

[21] and World Economic Forum [22]. External-facing staff also identified *external stakeholders'* keen desire to create connections to faculty expertise (8 M), to have a venue for talent identification (6 M), an interest in providing scholarships/grants, or to fund programs (5 M), and paying it forward by means of creating opportunities (3 M).

**External stakeholders' interest in STEM pre- and post-doctoral researchers identified by external-facing staff.** While STEM knowledge was discussed as important, business acumen and experience were deemed as the distinguishing features of a successful industry applicant (10 M). Creating an expertise or talent pipeline (10 M) motivates external companies to interact with external-facing offices at institutions. The mutual benefit to multiple stakeholders (10 M) and the desire to support student needs (4 M) are perceived to be the sustaining power of these relationships.

**External-facing staff's additional thoughts.** External-facing staff were eager to share suggestions for researchers to better prepare for their careers (11 M), such as being prepared to learn on the job independently, trusting critical thinking, and relying upon other transferrable skills developed in training. Based on their experience, external-facing staff provided suggestions regarding CPD programming, such as the need for new perspectives on training (6 M), ensuring one point of contact (4 M), and including an alumni engagement role (3 M).

For details, including representative comments for individual themes, see **S3 File**.

## Stakeholder 4: External partners–societies, foundations, non-profits

Interviews with societies, foundations, and non-profits indicated that there were many partnerships and exchanges of service through which they interact with academia including: networking and community building, providing resources for honing career skills, generating scholarship and publications, facilitating advocacy, providing feedback and advice, and creating funding opportunities.

Discussion with the stakeholders from societies, foundations and non-profits brought new themes to light. These highlight many available resources that are not always accessed by institutions. The most common reason societies, foundations, and non-profits provided for wanting to engage with academia was to build relationships to connect academics with the mission of their organization (**Fig 4**).

**External partner organizations' engagement with academia.** Building relationships (16 M) was a large motivator for societies, foundations and non-profits to interact with academia, as is the prestige, recognition, and public visibility (8 M) that comes with interacting with prestigious academic institutions. This engagement is also seen as a catalyst for connections and knowledge (5 M).

**Societies, foundations, and non-profits resources to offer.** Societies and foundations have several scientists on staff possessing a wide range of expertise and perspectives as a resource to offer (6 M), as well as varied online resources and guides (4 M), experiential learning opportunities (4 M), and self-exploration tools (2 M).

A few challenges faced by external partner organizations mentioned included limited funding (4 M), connecting with target audiences (3 M), and creating flexible, creative models along with the need for culture change (3 M).

**External partner organizations' view of career preparedness improvements.** The need for pre- and post-doctoral researchers to develop and broaden their skills and learning approach (11 M) was discussed frequently. Advice such as taking early initiative (8 M), improving self-efficacy and growth mindset (6 M), learning to listen with humanity and broaden diversity (4 M), making use of networking opportunities (2 M), and getting involved in professional societies (1 M) were provided, alongside the idea of defining one's own success (1 M).

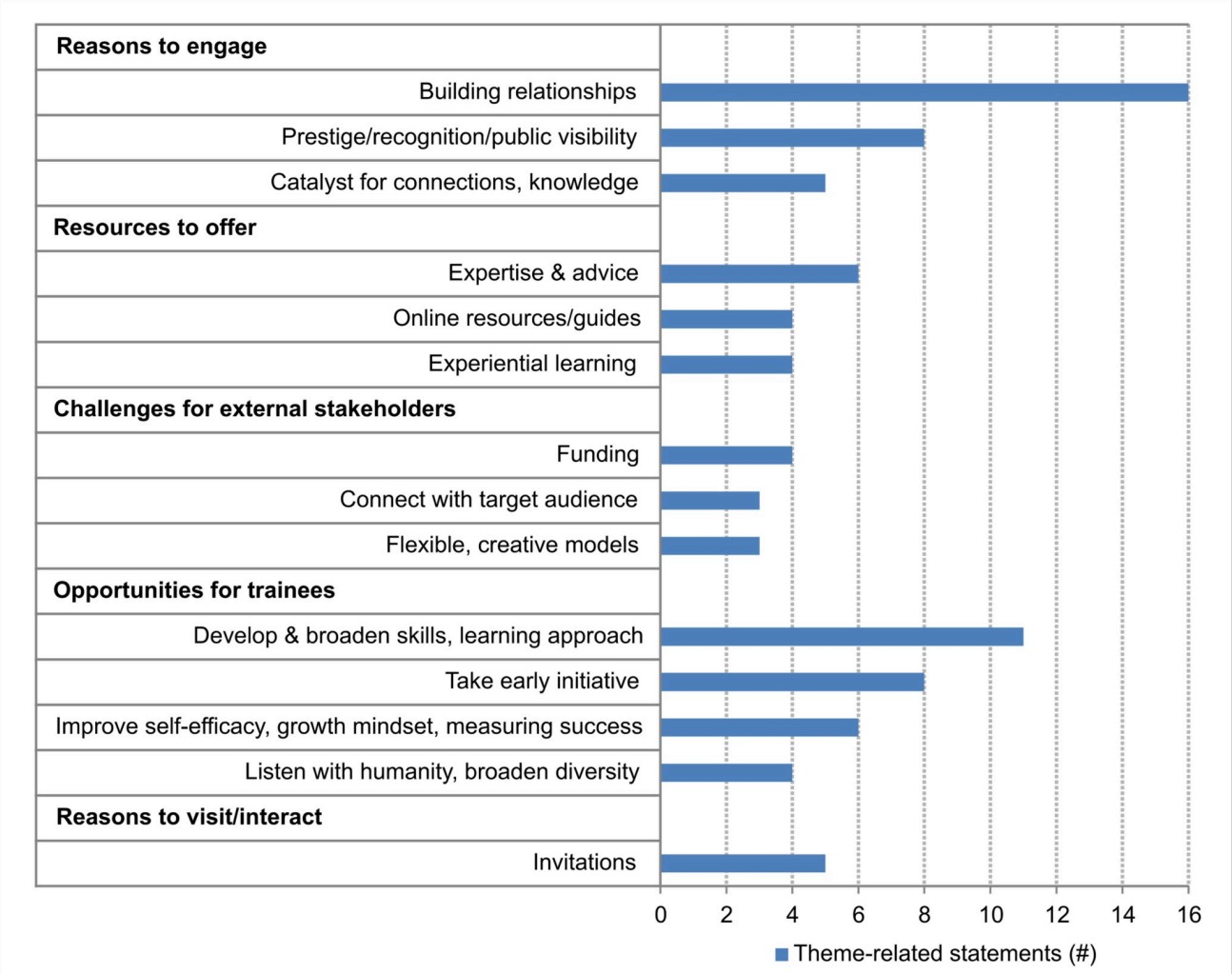

**Fig 4. Themes from external partner stakeholders: Societies, foundations, non-profits.** Bar graph representing the number of mentions of each theme for themes mentioned 3 or more times.

**External partners' view of engaging with academic institutions.** Non-profit and society stakeholders are keen to engage and disseminate their resources, and encourage institutions to coordinate on-campus visits (5 M), or organize for researchers to engage with them off campus (2 M). Some challenges discussed included the difficulty to integrate their offerings into training (2 M), their ability to engage at all levels with academic institutions (2 M), the perception that the wrong people make decisions regarding these partnerships (1 M), and the need for more industry mentors (1 M).

For details, including representative comments for individual themes, see **S3 File**.

## Stakeholder 5 –External employers–small and large companies, intellectual property firms, consultancies, accelerators

External employers interviewed for this study include representatives of large pharmaceutical, biotech, government or national labs, consulting firms, intellectual property firms, policy or

communication organizations as well as small business/start-ups, accelerators and boutique consulting agencies.

Types of engagement already in place with universities include: recruiting events, career fairs, tours/site visits, case studies/workshops, serving on advisory boards for curriculum development, and internships. A key goal for this kind of engagement is to maintain relationships.

This representative sampling of interviews with *external employer stakeholders* revealed additional themes and underscored themes already brought to light in the previous interviews (**Fig 5**).

**Reasons why industry external stakeholders engage with academia.**  Companies desire being involved in professional development programs, as it helps recruit and broaden their reach (8 M), and are keen to build long-term relationships (5M), often facilitated by an alumna/us acting as the liaison. There appears to be a need for bidirectional partnerships (4 M), and advisory and feedback roles (3 M).

**Training and collaboration industry-academia partnership benefits.**  The modes of interaction between industry and academia include alumni working with their *alma mater* (8 M). Increasing awareness of industry's resources to develop academia's complementary skills was discussed (6 M), as well as advice to pre-doctoral and post-doctoral researchers to learn industry-relevant skills (4 M), recalibrate what is considered important in job functions (3 M), and make use of grants and academic collaborations (3 M).

An important aside arose that if a company's needs are already met, they admitted to not seeking out interactions with academia.

**Differing priorities and organizational complexities–Challenges with industry.**  Challenges to industry interactions with academia include distrust due to perceived differing values (5 M), as well as a lack of a single point of contact at academic institutions (3 M).

**External industry employers' views on challenges with pre- and post-doctoral researcher preparedness.**  Understanding career options, industry culture, and priorities (8 M) are considered critical to being a successful industry applicant. The need for pre- and post-doctoral researchers to develop skills such as good communication (8 M), present experience and motivation appropriately to employers (7 M), and increase focus on relationship building and collaborations (5 M) are seen as crucial skills for success in industry. The need for faculty culture change (4 M) to improve CPD relationships with industry was also discussed.

**How to encourage industry professionals to interact and visit campus.**  Several companies are glad to visit or interact with academic institutions upon receiving invitations, ideally by pre-doctoral or post-doctoral researchers (6 M), while some prefer attending high-impact events (3 M), match-ups (3 M), or hosting site visits at the company (1 M). Industry representatives advise academic leaders to be open-minded to industry (3 M), and mention the challenge of limited time and resources to engage with academia (1 M).

The vast majority of large and small external employers interviewed were unaware of the national conversation about career development and outcomes of PhD-trained scientists.

## Stakeholder engagement tool purpose, use, and opportunities for action

A stakeholder engagement tool was created for practitioners to rapidly assess and identify with which stakeholder group they are primed to interact most efficiently. The tool can help practitioners quickly focus on existing strengths at their institution on which they can rely, as well as on areas of improvement and possible links to approach stakeholders strategically. Coupled with the themes found in the interview data, a targeted approach can be developed to improve stakeholder engagement.

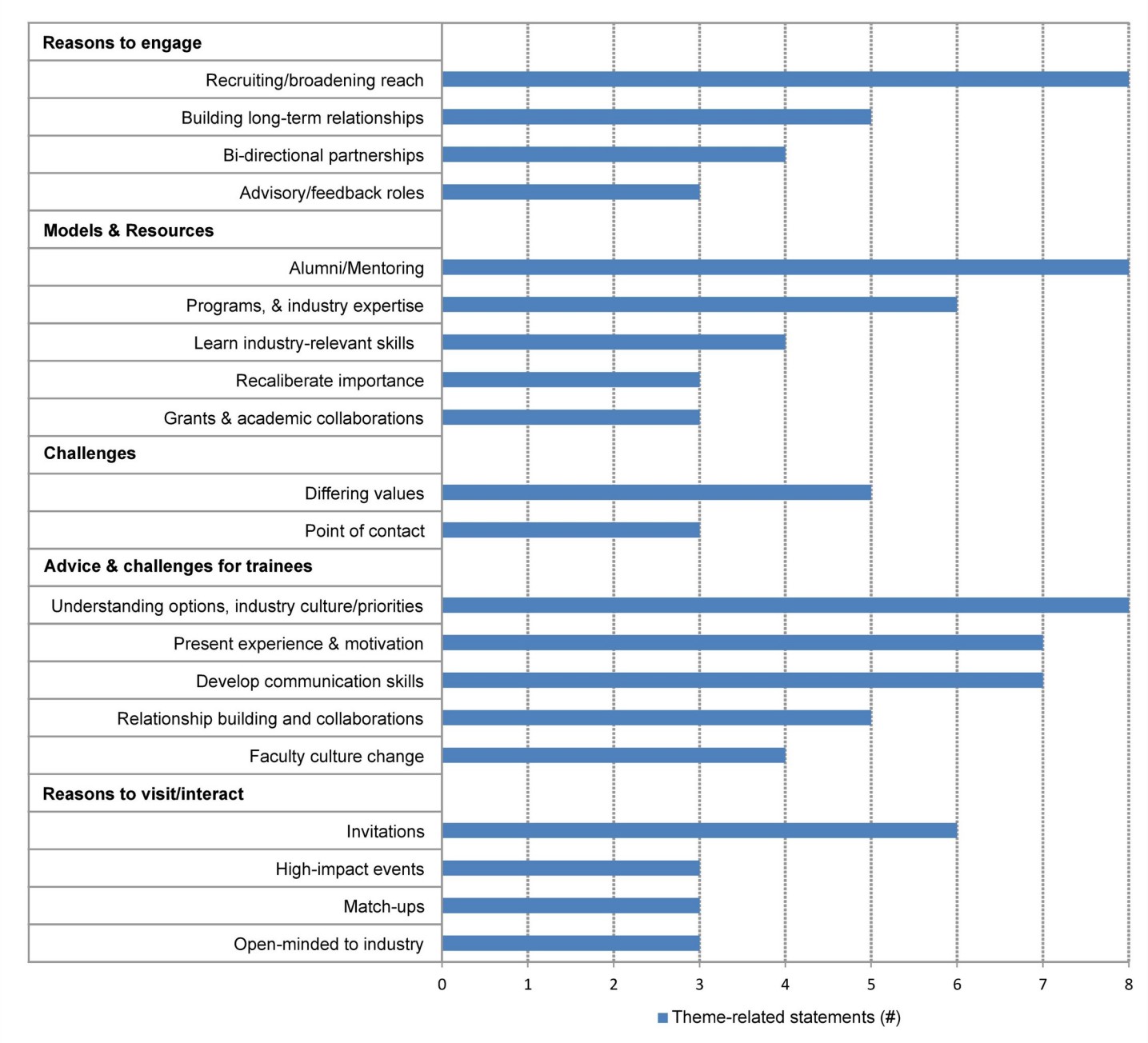

**Fig 5. Themes from external employer stakeholders–Industry–Large and small businesses.** Bar graph representing the number of mentions of each theme for themes mentioned 3 or more times.

The stakeholder engagement tool is fully customizable to reflect local organizations with whom to partner and those that already might have existing relationships. A 360-degree view of perceived stakeholder engagement can be quickly determined by encouraging colleagues around campus to fill out the tool. The resulting scores will inform discussion across offices to see where perceptions align and where there might be differences in scores. Since the tool is quick and easy to use and automatically creates a visual output by summing scores in each quadrant of the ensuing graph (representing internal pre- and post-doctoral researchers,

faculty/administrators; external-facing staff; external partners; and external employers) practitioners gain a quick, holistic view of their engagement.

To interpret the output of the stakeholder engagement tool, use the Excel file (**S5 File**; Note: Please download the Excel file to view proper format) which automatically sums each quadrant. The scores toward the left or right of the plot indicate areas of focus externally (right, blue and grey) or internally (left, orange and red). The top half of the plot reveals information on internal/external users (in orange and blue), the lower half on internal/external partners (in red and grey). The sums in each quadrant indicate the relative strength in each stakeholder group (top left: *internal stakeholders* such as graduate students or faculty; top right: *external stakeholders* such as employers; bottom left: *internal* partners such as licensing office; and bottom right: *external* partners such as professional societies).

This is a self-reflection tool to identify areas of individual network engagement and areas for potential development. The more the sectors are filled to the outside of the circle, the more **perceived** relative engagement there is. Based on how you define your Likert scale (see suggested scale in Methods), the score for the relative engagement may reflect representation of stakeholders belonging to each group, as well as frequency and quality of interactions between stakeholders. For example, the CPD office may only have one or two stakeholders in a given stakeholder group, but you might meet with them often and they might be extremely influential and enthusiastic about supporting CPD programs for graduate students and postdoctoral researchers. Moreover, it is important to engage with and address the concerns of naysayers, as this will add value to overall CPD operations and help to launch more successful programming.

In the example (**Fig 6**), the *internal stakeholders* are visually a strength, especially graduate students, and there appears to be room for increased engagement with certain *external stakeholders* such as intellectual property firms.

There are three basic approaches to action as a result of filling out the stakeholder engagement tool.

1. **Evaluate competencies.** Using this stakeholder engagement tool, start conversations in your group to identify a shared framework to address areas of strength and growth. Choose 1–3 areas to connect with stakeholders (some might be linked or be in one quadrant). Identify the decisions to be made and develop an action plan based on these linked competencies; for example, local companies and an accelerator/incubator might provide more opportunities around entrepreneurial career paths for postdocs.

2. **Identify challenge areas.** Look at the chart to identify three barriers to potential growth in stakeholder engagement (lowest scores, to the inside of the circle). Low scoring areas should be evaluated for feasibility and potential actions to take (or non-relevancy). These areas might form the basis for discussion of methods to overcome challenges. Pay attention to challenges pertaining to diversity among stakeholder networks, and reflection of stakeholder diversity as it pertains to the trainee population, such as international/domestic participants, gender or gender identities, and race/ethnicity. For example, conversations with your industry relations office, national labs and a trade organization might spark ideas; perhaps there are no business/tech parks nearby so a virtual site visit might increase interactions.

3. **Map your strengths.** In looking at the chart, identify three strengths (points reaching most to the outside of the circles). Look for strengths in each quadrant of users, internal/external partners, and employers. Identify key individuals in these areas and bring them together with your team to discuss next steps for engagement; for example, pre-doctoral researchers, career services and professional society representatives might meet to brainstorm ideas.

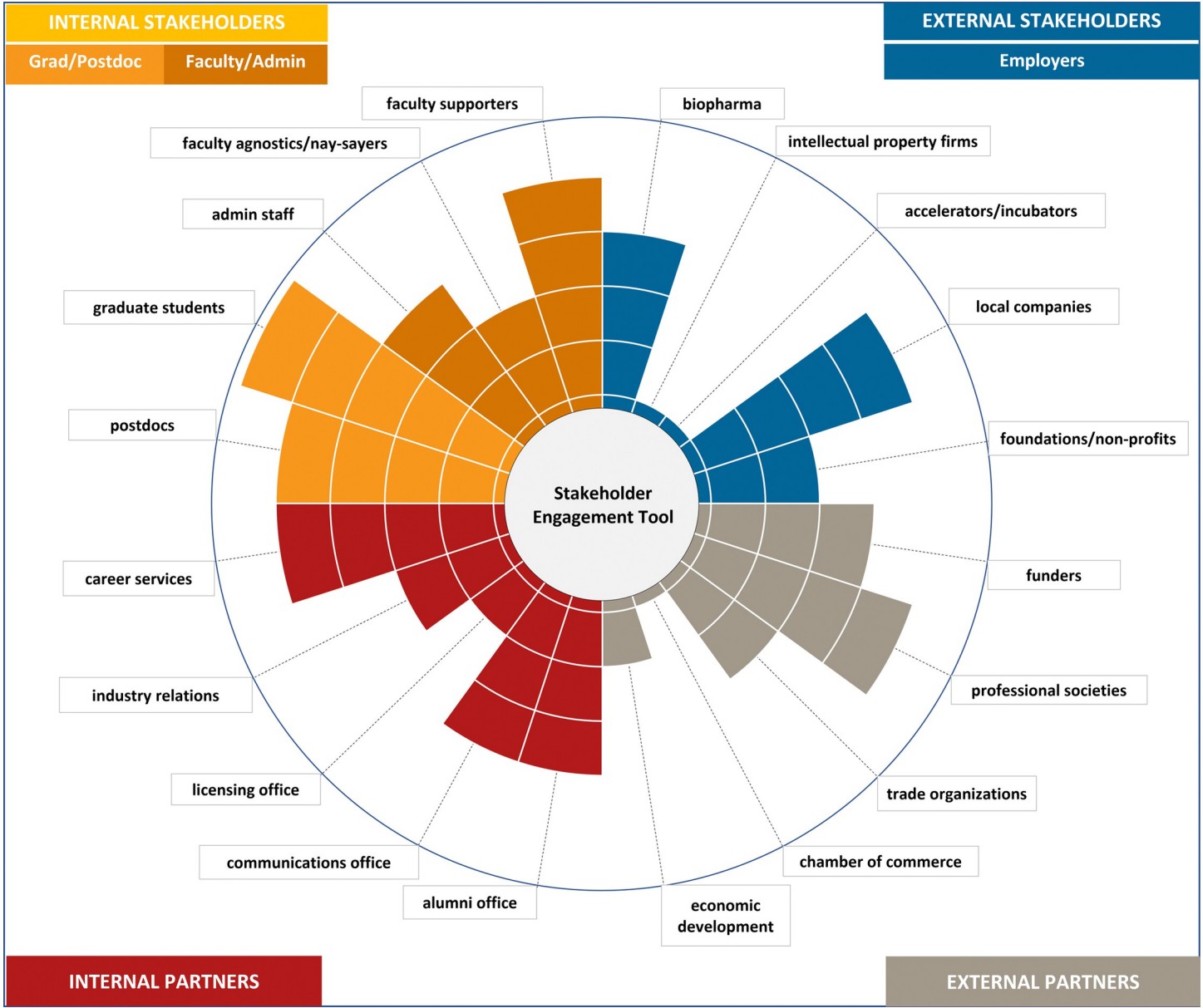

**Fig 6. Stakeholder engagement tool.** A rapid assessment tool for *internal* and *external stakeholders* to evaluate competencies and determine strengths for engagement in career and professional development programming. Detailed instructions for entering values are in Methods, and interpretation can be found in Results. [see S5 File to download and use the stakeholder engagement tool and view proper format: https://osf.io/fc27x/?view_only=b166987514234b718d8457778651534f].

## Discussion

### Common themes

When considering the diverse stakeholder groups, common themes emerged across the different internal and external groups. Themes that reappeared across the various stakeholders suggest that these topics should be priorities for CPD practitioners when attempting to engage broadly.

Individuals from all interviewed stakeholder groups recognized the value of CPD activities for pre- and post-doctoral researchers, despite different perspectives on similar concepts, with individual participants varying in levels of enthusiasm and commitment. For instance, there is

a distinct need for CPD programs across stakeholder groups, who engage for varying reasons. Among *internal stakeholders*, most pre- and post-doctoral researchers believe these activities benefit their career growth, many faculty and administrators believe the programs strengthen researcher career development and benefit their mental health, and all external-facing offices are keenly aware of both the value to pre- and post-doctoral researchers and how these activities generate interest among *external stakeholders*. External partners and employers desire that researchers are well-prepared when entering the workforce, and encourage CPD programs to partner with them on program development. Notably, external partners develop many resources to aid CPD activities for pre- and post-doctoral researchers and encourage them to be involved and to use these resources.

Networking, making connections, partnering, and collaborating are seen as crucial aspects of CPD across all stakeholder groups. Pre- and post-doctoral researchers are interested in opportunities to build and grow their networks to help form their future careers, and faculty/administrators understand this to be critical to researcher development. Non-profits and foundations encourage non-prescriptive models for graduate learning, which could open the doors for more engagement across stakeholders, and industry partners offer assistance in developing programming and internship opportunities. Financial arrangements can be mutually beneficial: universities can benefit financially from industry interactions to support their research, and collaborative commercialization of research benefits industry. Additionally, *external stakeholders* are interested in connecting with academic institutions to have early access to emerging science and technology, develop partnerships to grow their own priorities, and build a talent pipeline. External employers also showed keen interest in collaborating with faculty, and recommend that academics engage with industry partners on joint publications to help highlight these partnerships in the media [23], and facilitate culture change regarding opinions on industry collaborations.

The timing and content of CPD activities was another important focal point with multiple subgroups (pre- and post-doctoral researchers, external-facing staff, external partners and employers) suggesting the need for integrated, persistent, embedded, and flexible access to these activities. In particular, pre- and post-doctoral researchers suggest that there is a need for consistent exposure to CPD throughout training, including by some infrequent users who believe that this programming should be woven into the curriculum for maximal benefit, as is common in professional schools. Historically, the common opinion was that pre- and post-doctoral researchers should focus on their careers *after* they complete their training, but this is not ideal as it delays the workforce pipeline [9]. In addition, some faculty stakeholders do not see the value of CPD activities and expressed some concern about the time their pre- and post-doctoral researchers dedicate to these activities. However, recent evidence-based research has shown that participation in internships, career development programming, K-12 outreach programs or IRACDA programs does *not* lead to increased time to degree or decreased productivity [3, 24–26]. Pre- and post-doctoral researchers, faculty/administrators and external partners all noted the value of flexible programming to encourage pre- and post-doctoral researcher engagement. All stakeholders pointed to the need to remain informed about the needs of both researchers and the workforce when planning, designing, and executing CPD activities.

An important challenge identified by pre- and post-doctoral researchers, external-facing partners, and *external stakeholders* was the need to expand the purview of scientific training to include skill development for a variety of careers. Alongside this, these stakeholders comment on the challenge of normalizing CPD activities in the larger context of training, and the need to help academic leaders (e.g. faculty and administrators) understand that academic career preparation is only a part of CPD, and that other types of skill development are necessary, as they are complementary and important to the success of their pre- and post-doctoral

researchers. For example, developing self-efficacy is extremely beneficial to researchers when approaching their CPD [2, 27]. Relatedly, requiring faculty approval to participate in professional development activities can present a barrier to participation and reinforces the cultural stereotype that professional development is outside of the normal expected activities of a pre- and post-doctoral researcher. Despite federal agencies [28], clarifying that pre- and post-doctoral researchers' skill development is critical, the concept of career exploration has not permeated to all faculty and administrators–it is evident that for successful CPD implementation, practitioners need to be active in outreach and engagement with internal faculty [5, 9] and administrators. CPD offices should work to identify new strategies for conveying their services and value to develop faculty/administrator buy-in, e.g. more evidence-based research to convey the program's benefits. Knowing the culture of local faculty and their attitudes toward CPD can help strategically design programs that will yield the highest number of participants [9].

Among other challenges discussed, one key challenge noted was the identification and implementation of streamlined methods to access or connect with the right resources or people. Pre- and post-doctoral researchers report struggling to identify resources at their institutions, suggesting the need for a centralized institutional CPD hub, while external-facing staff commented on not knowing the appropriate people within academic institutions with whom to connect their external contacts. *External stakeholders* recommend universities have a visible "one-stop shop", to encourage external partners or employers to connect with them. Additionally, *external stakeholders* note that it is critical for a graduate career office to have strong engagement between past and present CPD practitioners, to ensure continuity of relationships with the various stakeholders.

A disconnect was revealed specifically for *external stakeholders* who are unaware of the national conversation about CPD activities, suggesting that practitioners at academic institutions should more frequently intermingle in industry settings and more broadly disseminate their findings to *external stakeholders*, perhaps at industry conferences.

The authors' knowledge of *internal stakeholders* allowed them to engage with a spectrum of users of CPD services including frequent users, occasional users and non-users, as well as a range of faculty/administrators showing enthusiastic, cautious or no support for CPD activities. The variety of *internal stakeholders* interviewed resulted in valuable conversations to identify where CPD practitioners can improve. For example, pre- and post-doctoral respondent interviews cited requiring an increased awareness of their needs and purpose for engagement to better align existing CPD opportunities and guide new ones, while faculty/administrator interviews highlighted perceptions of CPD offices, identified concerns, collected suggestions on tailored experiences and exposures, and identified the need for clearly defined CPD. It appears valuable for CPD practitioners to have a clear understanding of *internal stakeholders'* needs and concerns to create effective programming. Simultaneously, university staff who engage with *external stakeholders* share similar interests to CPD offices that include supporting and giving advice to pre- and post-doctoral researchers. Hence collaborations with these partners can provide valuable *external stakeholder* perspectives.

In summary, common themes across all stakeholders are shown in **S6 File**. Many emerging common themes centered on a tailored approach to CPD programs. For example, while most stakeholders acknowledged the need for CPD, the desired timing and content of programming aligned with individual specific needs. This further accentuates the need for creating streamlined access, discussed by most stakeholders.

## Stakeholder engagement

The stakeholder engagement tool was debuted at an international meeting of CPD practitioners to demonstrate how to evaluate and plot to extend their networks in a targeted and structured way. Feedback from university-based users included surprise to learn the number of partners that could be leveraged by sometimes small, understaffed offices tasked with serving large populations of graduate students and postdocs. Subsequently, the tool's value was expanded to address diversifying not only the roles, sectors, and stakeholder groups existing within one's network, but also who and which social identities or wide varieties of lived experiences were represented among those contacts. Newly identified partnerships across campus as well as in the local community were seen as options not previously considered, including (but not limited to; see **S4 File** for more examples) partnerships with Historically Black Colleges and Universities (HBCUs), Minority Serving Institutions (MSIs), as well as Black, Indigenous and People of Color (BIPOC), Latinx and other groups, serving those who have been historically excluded and underrepresented in science (e.g., persons excluded due to ethnicity or race [29]). Benefits cited by meeting attendees included the rapid ability to identify a wide variety of stakeholders with whom to work and partners to increase opportunities for their pre- and post-doctoral scholars.

An added value identified by participants was the quick analysis of potential barriers to growth in stakeholder engagement that could guide future discussions to overcome challenges. The tool was also cited as useful because it was designed for customization to each institutional setting, e.g. whether appropriate for recruiting industry partners in the local area if high density of contacts is available regionally, or fitting to explore ways to connect across a university that is a large, decentralized behemoth. Templates for how to reach out to potential partners were also reported to be useful. Of course, the tool is intended only as a first step in engaging stakeholders. More research and time investment is needed to determine the exact person to reach out to at various organizations if no existing partnerships exist, but the process is intentionally step-wise so that over time more stakeholders can be involved in CPD to benefit all parties. The tool serves as a way to focus on strengths of existing relationships, hone in on partners to include in discussions, or optimize outreach to spark new collaborations across stakeholders, rather than to identify specific challenge areas or strengths within a stakeholder group. We would advocate to first use the tool as a preliminary screen to identify which stakeholder categories to focus on. Informed by these themes identified through our interviews, CPD offices can then follow up with stakeholders to develop strong partnerships. Should the stakeholder engagement tool indicate that increased interactions with faculty or university administrators is warranted, that might stimulate conversations among CPD offices, department chairs and deans, to better inform faculty of the advantage of transferrable skills for all careers. These actions will help promote culture change in academia and reinforce awareness of the range of career options beyond academia available to PhDs.

CPD practitioners should consider engaging with both alumni and future employers as key stakeholders. Alumni are one of the most accessible *external stakeholders* for graduate career development work. The personal experience of alumni in the workforce provides critical input for assessing skills required by employers and to inform curriculum changes [2, 9, 12]. Furthermore, alumni connections are valuable for establishing external partnerships. Individual institutions vary in their ability to cultivate and engage alumni, influenced greatly by the existence of an alumni relations office with active engagement strategies, for example, social media sites and accessible directories [10, 11].

Finally, the process of identifying *external employer stakeholders* as well as engagement opportunities should include strategic consideration of the location of the institution. For

example, universities located in urban areas with a high concentration of biopharma companies might develop mechanisms to promote pre- and post-doctoral researchers' biomedical expertise, valuable to local *external stakeholders* [10, 30]. More isolated universities might organize a conference or trek to a more biotech-rich area [10, 23].

### Limitations and future directions

Faculty and administrators were sampled from a wide variety of roles, ranging from rank-and-file faculty; through departmental leadership such as Chairs, Associate Deans, and other leaders; through leaders in graduate and postdoctoral education such as Deans of Students. Nonetheless, we acknowledge the importance of identifying unique themes and perspectives that may arise for Provosts, Presidents, Chancellors, and other high-level university-wide leadership roles which are crucial to setting strategic aims, funding, and program priorities. The current sample did not include a large enough subset of high-level leaders to analyze the data separately, and future directions of research could include examining this group of administrators specifically.

*External stakeholders* were sampled from a variety of roles in industry and non-profit employers including small and large companies, intellectual property firms, consultancies, and accelerators. While this study did not specifically sample alumni as a stand-alone stakeholder group, some *external stakeholder* individuals (e.g., For-Profit, Non-Profit/Society) were incidentally alumni of the institutions represented. Still, we realize it is important to include the voice of recent PhD and postdoctoral alumni for their understanding of how their training affected their employability, how their industry sector views PhDs, and their retrospective views of their careers. This is especially of interest since they are role models for our current students and postdoctoral researchers, and are now potentially in hiring positions. Alumni are a very strong stakeholder group to engage with, as they are eager to give back to their institution, are invested in the institution, and directly experienced programs and the various stakeholders involved in the programs. Hence, investigating this stakeholder group would be extremely useful as a future research topic.

Further analyses that merit additional attention include examining unique themes arising through intersectional identity groups (e.g., gender, race, ethnicity, international/domestic status, among others). Unfortunately, sample sizes within each stakeholder group were not large enough to examine these important differences; however, this is an important topic for additional work to consider.

Another useful future analysis would be to ask stakeholders to rank which challenges represent the biggest barriers, a function that was not queried in this study's interviews. Such a ranking would assist CPD offices to prioritize resources to overcome challenges.

Though our research interviewed only people currently residing in the U.S., the importance of CPD globally should not be understated. A recent report from the Organisation for Economic Cooperation and Development (OECD) representing a coalition of 38 nations also emphasized the need for tracking PhD alumni career outcomes to help understand where the programs can increase their CPD, as postdoctoral researchers are outgrowing the number of tenured or permanently employed academic positions in many countries [31]. The importance of CPD and culture change around CPD is critical not only for the U.S. but also globally, and future research could garner more insights from a global perspective, although we recognize that the one-size fits all approach wouldn't apply to all countries.

### Conclusions

This study brings to light fundamental career and professional development (CPD) concepts that span the various *internal* and *external stakeholder* groups interviewed. Learning from

these opinions is valuable, and can help form recommendations in the creation, design, and sustenance of effective CPD activities at individual institutions.

This study also presents the stakeholder engagement tool, which can be used for rapid self-analysis of practitioners' networks to assess strong stakeholder relationships and areas where the practitioner can strengthen their network. Coupled with the various themes from interviews with 45 *internal* and *external stakeholders* across the country in various roles, graduate career practitioners can use the themes presented as discussion points to interact with their own stakeholders to prepare for potential meetings with new contacts. Meaningful and targeted engagement with various stakeholders is key to create and sustain successful graduate and postdoctoral CPD programs.

## Supporting information

**S1 Table. Number of interviews conducted per stakeholder subgroup, by interviewer.**
(PDF)

**S2 Table. Pseudonym assignments for stakeholder interviews.**
(PDF)

**S1 File. Rationale and sample questions for stakeholders.**
(PDF)

**S2 File. Sample wording for invitation to participate.**
(PDF)

**S3 File. Detailed results, including example comments.**
(PDF)

**S4 File. Examples for diversifying networks.**
(PDF)

**S5 File. Stakeholder engagement tool.**
(XLSX)

**S6 File. Common themes across stakeholders.**
(PDF)

## Author Contributions

**Conceptualization:** Deepti Ramadoss, Amanda F. Bolgioni, Rebekah L. Layton, Janet Alder, Natalie Lundsteen, C. Abigail Stayart, Jodi B. Yellin, Susi S. Varvayanis.

**Data curation:** Deepti Ramadoss, Amanda F. Bolgioni, Rebekah L. Layton, Janet Alder, Susi S. Varvayanis.

**Formal analysis:** Deepti Ramadoss, Amanda F. Bolgioni, Janet Alder, Susi S. Varvayanis.

**Funding acquisition:** Deepti Ramadoss, Amanda F. Bolgioni, Rebekah L. Layton, Janet Alder, Susi S. Varvayanis.

**Investigation:** Deepti Ramadoss, Amanda F. Bolgioni, Rebekah L. Layton, Janet Alder, Susi S. Varvayanis.

**Methodology:** Amanda F. Bolgioni, Rebekah L. Layton, Natalie Lundsteen, C. Abigail Stayart, Susi S. Varvayanis.

**Project administration:** Deepti Ramadoss, Amanda F. Bolgioni, Rebekah L. Layton.

**Resources:** Deepti Ramadoss, Amanda F. Bolgioni, Rebekah L. Layton, Janet Alder, Susi S. Varvayanis.

**Software:** Conrad L. Smart, Susi S. Varvayanis.

**Supervision:** Deepti Ramadoss, Amanda F. Bolgioni, Rebekah L. Layton, C. Abigail Stayart.

**Validation:** Deepti Ramadoss, Amanda F. Bolgioni, Rebekah L. Layton, Janet Alder, Natalie Lundsteen, C. Abigail Stayart, Jodi B. Yellin.

**Visualization:** Deepti Ramadoss, Conrad L. Smart, Susi S. Varvayanis.

**Writing – original draft:** Deepti Ramadoss, Amanda F. Bolgioni, Rebekah L. Layton, Janet Alder, Natalie Lundsteen, Susi S. Varvayanis.

**Writing – review & editing:** Deepti Ramadoss, Amanda F. Bolgioni, Rebekah L. Layton, Janet Alder, Natalie Lundsteen, C. Abigail Stayart, Jodi B. Yellin, Conrad L. Smart, Susi S. Varvayanis.

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
