## [Decision Letter · Decision Letter 0]

5 Oct 2021

PONE-D-21-26337Using Stakeholder Insights to Enhance Engagement in PhD Professional DevelopmentPLOS ONE

Dear Dr. Ramadoss,

Thank you for submitting your manuscript to PLOS ONE. After careful consideration, we feel that it has merit but does not fully meet PLOS ONE’s publication criteria as it currently stands. Therefore, we invite you to submit a revised version of the manuscript that addresses the points raised during the review process. Your manuscript addresses an important yet underserved topic in higher education. Overall, the manuscript is very well written, and methods and limitations are appropriately described. However, and as mentioned by reviewers, minor revisions, particularly in describing the tools/graphs, need to be made to improve this submission. We look forward to receiving your resubmission.

We look forward to receiving your revised manuscript.

Kind regards,

Sina Safayi, D.V.M., Ph.D.

Academic Editor

PLOS ONE

Journal Requirements:

3. Please ensure that you refer to Figure 1-6 in your text as, if accepted, production will need this reference to link the reader to the figure.

Reviewers' comments:

Reviewer's Responses to Questions

**Comments to the Author**

1. Is the manuscript technically sound, and do the data support the conclusions?

Reviewer #1: Yes

Reviewer #2: Yes

Reviewer #3: Yes

2. Has the statistical analysis been performed appropriately and rigorously? 

Reviewer #1: Yes

Reviewer #2: N/A

Reviewer #3: Yes

3. Have the authors made all data underlying the findings in their manuscript fully available?

Reviewer #1: No

Reviewer #2: Yes

Reviewer #3: Yes

4. Is the manuscript presented in an intelligible fashion and written in standard English?

Reviewer #1: Yes

Reviewer #2: Yes

Reviewer #3: Yes

5. Review Comments to the Author

Reviewer #1: Congratulations to the authors on an important and timely paper. This research was very well designed and written, and the paper was a pleasure to read. The figures were well developed and informative and the interpretation of data was meaningful.

While not material to accepting this article I do have a couple of questions and comments which I think would strengthen your data or should be considered:

1. The inclusion of PhD graduates 5-10 years out as a stakeholder group (i.e. not postdocs) from different career paths would have been a valuable data source. Those working in industry could have provided some interesting responses regarding the value of CPD in their workplace, how their training affected their employability, the view of their colleagues/ industry on PhDs and the choices they would have made in retrospect. Those still in academia, similarly would have had interesting responses that perhaps would be slightly different to the senior academics interviewed regarding career needs and training.

2. Despite the pandemic, global mobility of research graduates is an expectation and as such it would have been good to see more of a recognition of the "international" conversations about career development and outcomes of PhD and the international competitiveness of universities globally which are offering well developed CPD in their programs. The implementation of CPD into PhD programs has global momentum and this is beginning to influence student recruitment and industry partners wishing to employ graduates; so it is important to recognise this.

3. As the data suggests, there is still a need for both education and culture change in faculties for academic advisors to understand the idea that career paths are very different from their graduate days, that the concept of leaving academia forever is no longer as binary and that CPD in career and transferable is essential for both academic and industry careers It would have been great to touch on this more about how this could also be addressed by universities (and is it being done already?) so that the use of your tool could be better implemented by these stakeholders.

Overall, a lovely paper, thank you for your submission.

Reviewer #2: PONE-D-21-26337, titled "Using Stakeholder Insights to Enhance Engagement in PhD Professional Development" by Ramadoss et al., outlines the benefits of identifying and engaging different groups of internal and external stakeholders in developing, institutionalizing and communicating value of career and professional development (CPD) for STEM pre-doc and postdoc scholars. In this article, authors develop and utilize a stakeholder engagement visualization tool to inform strategy, priorities and approaches to action for CPD practitioners. As highlighted by the authors, identifying different stakeholders, understanding incentives of engagement and priorities of each is important especially for practitioners in small offices with fewer resources. Also, the attention given to perspectives of stakeholders from diverse demographic backgrounds, who may belong to underrepresented and historically excluded groups is noteworthy. The article is valuable for CPD practitioners in a field that’s growing fast and would benefit from consistent approaches and best practices. I recommend approval with following suggestions and considerations for revisions and improvement.

Major comments

1. Narrative: The authors primary audience are CPD practitioners and their internal stakeholders. According to me, the narrative needs to balance two primary goals: 1. communicating importance of stakeholder engagement to inform and direct CPD efforts, and 2. benefits of utilizing the stakeholder engagement visualization tool. The authors need to better outline the second goal. The result section communicates the importance of stakeholder engagement to inform and direct CPD efforts through text analysis of interview responses. However, the visualization tool itself needs more introduction and context within the text (either in methods or results). Figure 6 was confusing for me. I wish the instructions within the figure 6 were embedded in the text of the article for the reader. The output of the visual seems to corelate to number of stakeholders in each groups and level of engagement. However, I was confused by mention of topics and quadrants. When I hovered on parts of the visual, I could see “A” or “S” but didn’t know what topics “A or S” indicate.

2. Perspectives of institutional leaders as key internal stakeholders: Authors mention graduate deans and provosts are potential stakeholders for consideration. The study would benefit from including graduate deans’ perspective. This is especially important as equitable access and institutionalizing CPD comes up multiple times as a theme. While faculty chairs perspective is important, graduate deans have the agency to develop structures and policies, as well as allocate resources to make CPD more accessible and embedded within graduate programs. Their perspective of CPD is a key piece of this puzzle. Similarly, perspectives of representatives from Vice President for Research and/ or Provost division is important for institutionalizing CPD for postdoctoral training for equitable access beyond training grant recepients.

3. Approaches to action: In the discussion section, authors highlight the benefits of this engagement resource and utilizing and customizing the tool to create approaches to action. They also summarize some shared themes and incentives across stakeholder groups. To strengthen that point, I would recommend creating a figure with graphic representation of shared themes and ranked barriers or challenges.

Minor comments

4. Specific guidance and considerations for URM perspectives: Authors indicate percent of interviewees who belong to different racial-ethnic demographic and international population. Given the intersectional identities, could authors draw how many of the interview participants would identify as “under-represented” in their field of study or employment and how their perspectives may provide dimensions to strengths and challenges of CPD? I bring this up as authors provide a good list of identities to consider in each stakeholder group. However, how should CPD practitioners utilize those perspectives to examine nuances to balance with broad priorities. E.g. one interview participant highlights importance of CPD in recruitment of talented graduate students. Studies indicate that CPD is particularly important in recruitment, retention, success and wellbeing of early career scientists from under-represented and historically excluded groups. I wonder whether there’s a scope to analyze the data from this lens after aggregation and making specific recommendations and considerations in ‘approaches to action’. Offering specific guidance on approaching overlapping and distinct themes emerging from perspectives of under-represented and historically excluded stakeholder groups will further highlight the importance of this approach and tool for diverse stakeholders in graduate education, beyond CPD practitioners.

5. How are alums positioned within internal and external stakeholder populations? Authors focused on distinct sets of interview questions for internal and external stakeholders. However, alumni are an interesting population and may provide key perspectives for both sets of questions. Authors could recommend a customized hybrid interview questionnaire for alumni which draws from both internal and external stakeholder interviews. In fact, the alumni survey questions in CGS PhD Career Pathways Project presents a similar hybrid scenario spanning retrospective reflection as a PhD graduate and perspective as a professional.

Reviewer #3: This paper details a sound methodology for engaging with stakeholders in researcher career development that aims to assist graduate career educators to navigate relevant networks internal and external to their academic institution. By engaging internal-facing networks the methodology allows possible supporters and non-supporters of the career and professional development of researchers within an academic setting to be identified and challenges and opportunities to be identified. The external facing internal stakeholders and external stakeholders all provided feedback on the need for CPD in order to facilitate the future career development of pre- and postdoctoral researchers outside of academia.

In contrast with the very complete description of the stakeholder engagement methodology, the stakeholder visual engagement visualization tool is very scantily described. The authors claim that the tool "is quick and easy to use" without revealing exactly how the visualized data is collected. For example, it is not clear if the scores relate to the number of stakeholder belonging to each group or the quality of their comments or the quality of their support for CPD or of the ease that graduate career educators can access stakeholders from the relevant groups. In Figure 6 "Faculty agnostics/nay-sayers" have a score higher than "faculty supporters", which seems unlikely for quality based responses. Also it appears that "Faculty supporters" is given the color for the wrong quadrant "Employers". Without a description of what the scores represent, it is not clear how the rapid tool can be used to identify challenge areas and strengths. As only stakeholder groups are scored, and not specific challenges and strengths, it is hard to see how this stakeholder visual engagement visualization tool can compete with the time-intensive process described in the rest of the paper that delivers exactly those results. I recommend that the authors resubmit the paper with an improved description of the visualization tool.

6. PLOS authors have the option to publish the peer review history of their article (what does this mean?). If published, this will include your full peer review and any attached files.

Reviewer #1: No

Reviewer #2: No

Reviewer #3: No

---

## [Author Response · Author response to Decision Letter 0]

19 Nov 2021

Editor’s Comments:

E1. Thank you for submitting your manuscript to PLOS ONE. After careful consideration, we feel that it has merit but does not fully meet PLOS ONE’s publication criteria as it currently stands. Therefore, we invite you to submit a revised version of the manuscript that addresses the points raised during the review process.

Your manuscript addresses an important yet underserved topic in higher education. Overall, the manuscript is very well written, and methods and limitations are appropriately described. However, and as mentioned by reviewers, minor revisions, particularly in describing the tools/graphs, need to be made to improve this submission. We look forward to receiving your resubmission.

EA1. We thank you for your feedback, and have addressed the revisions requested, described below.

Journal Requirements:

E1.1. Please ensure that your manuscript meets PLOS ONE's style requirements, including those for file naming. The PLOS ONE style templates can be found at 

EA1.1. Thank you for these templates; we have formatted the manuscript as per PLOS ONE style templates.

E1.2. In your Data Availability statement, you have not specified where the minimal data set underlying the results described in your manuscript can be found. PLOS defines a study's minimal data set as the underlying data used to reach the conclusions drawn in the manuscript and any additional data required to replicate the reported study findings in their entirety. All PLOS journals require that the minimal data set be made fully available. For more information about our data policy, please see http://journals.plos.org/plosone/s/data-availability.

Upon re-submitting your revised manuscript, please upload your study’s minimal underlying data set as either Supporting Information files or to a stable, public repository and include the relevant URLs, DOIs, or accession numbers within your revised cover letter. For a list of acceptable repositories, please see

http://journals.plos.org/plosone/s/data-availability#loc-recommended-repositories. Any potentially identifying patient information must be fully anonymized.

EA1.2. Thank you for pointing this out. Relevant underlying data are available in our OSF folder. We have clarified this in our updated Data Availability statement below. 

Minimal underlying de-identified data is presented in the results and figures. Additional clarifying de-identified data, in accordance with IRB approved protocols and in accordance with PLOS guidelines for qualitative data, is available as Supplemental Information (S3 File, S6 File) available from the Open Science Framework (OSF) database (https://osf.io/fc27x/?view_only=b166987514234b718d8457778651534f)

E1.3. Please ensure that you refer to Figure 1-6 in your text as, if accepted, production will need this reference to link the reader to the figure.

EA1.3. Thank you for this instruction: we have now included references to Figures 1-6 in the text. 

E1.4. Please review your reference list to ensure that it is complete and correct. If you have cited papers that have been retracted, please include the rationale for doing so in the manuscript text, or remove these references and replace them with relevant current references. Any changes to the reference list should be mentioned in the rebuttal letter that accompanies your revised manuscript. If you need to cite a retracted article, indicate the article’s retracted status in the References list and also include a citation and full reference for the retraction notice.

EA1.4. Thank you. As far as we are aware, none of the references in our reference list have been retracted. However, per review, we have included one new reference, and edited citations of two references. 

The new reference was included to address a comment raised by Reviewer#1 in R1.2: OECD. Reducing the precarity of academic research careers. OECD Science, Technology and Industry Policy Papers. 2021.

Two edited references include: 

[3] Brandt et al. 2021 – we had originally cited the preprint paper, which has subsequently been published in a peer-reviewed journal.

[15] Blumer 1954 – as the citation was not appropriately formatted.

Reviewers' comments:

Reviewer #1: 

R1. Congratulations to the authors on an important and timely paper. This research was very well designed and written, and the paper was a pleasure to read. The figures were well developed and informative and the interpretation of data was meaningful.

RA1. Thank you very much. We are glad to hear that our research topic resonates with you, and that the figures and interpretation were valuable.

R1.1. While not material to accepting this article I do have a couple of questions and comments which I think would strengthen your data or should be considered:

1. The inclusion of PhD graduates 5-10 years out as a stakeholder group (i.e. not postdocs) from different career paths would have been a valuable data source. Those working in industry could have provided some interesting responses regarding the value of CPD in their workplace, how their training affected their employability, the view of their colleagues/ industry on PhDs and the choices they would have made in retrospect. Those still in academia, similarly would have had interesting responses that perhaps would be slightly different to the senior academics interviewed regarding career needs and training.

RA1.1. Thank you for your insightful comments. Though some of the external stakeholder group members were alumni of the authors’ respective universities we did not intentionally approach the group of alumni PhD graduates who are 5-10 years out for their insights. Reaching out to this population for their insights on how their training affected their employability, the view of their colleagues/industry on PhDs, and the choices they would have made in retrospect would be extremely insightful for our next paper! We have added your suggestion into our newly added “Limitations and Future Directions'' section of the Discussion:

“External stakeholders were sampled from a variety of roles in industry and non-profit employers including small and large companies, intellectual property firms, consultancies, and accelerators. While this study did not specifically sample alumni as a stand-alone stakeholder group, some external stakeholder individuals (e.g., For-Profit, Non-Profit/Society) were incidentally alumni of the institutions represented. Still, we realize it is important to include the voice of recent PhD and postdoctoral alumni for their understanding of how their training affected their employability, how their industry sector views PhDs, and their retrospective views of their careers. This is especially of interest since they are role models for our current students and postdoctoral researchers, and are now potentially in hiring positions. Alumni are a very strong stakeholder group to engage with, as they are eager to give back to their institution, are invested in the institution, and directly experienced programs and the various stakeholders involved in the programs. Hence, investigating this stakeholder group would be extremely useful as a future research topic.”

R1.2. Despite the pandemic, global mobility of research graduates is an expectation and as such it would have been good to see more of a recognition of the "international" conversations about career development and outcomes of PhD and the international competitiveness of universities globally which are offering well developed CPD in their programs. The implementation of CPD into PhD programs has global momentum and this is beginning to influence student recruitment and industry partners wishing to employ graduates; so it is important to recognise this.

RA1.2. We appreciate your insights on the inclusion of international career and professional development into this manuscript. Though all of our interviewees currently reside within the U.S., a number of them received their PhDs from outside of the U.S. and some mentioned the career and professional development programs in countries where they received their PhD training. We recognize the importance of including international PhD CPD within the manuscript and have added your insights into our newly added “Limitations and Future Directions'' section of the discussion:

“Though our research interviewed only people currently residing in the U.S., the importance of CPD globally should not be understated. A recent report from the Organisation for Economic Cooperation and Development (OECD) representing a coalition of 38 nations also emphasized the need for tracking PhD alumni career outcomes to help understand where the programs can increase their CPD, as postdoctoral researchers are outgrowing the number of tenured or permanently employed academic positions in many countries [31]. The importance of CPD and culture change around CPD is critical not only for the U.S. but also globally, and future research could garner more insights from a global perspective, although we recognize that the one-size fits all approach wouldn’t apply to all countries.” 

[31] OECD (2021), OECD Science, Technology and Industry Policy Papers: Reducing the precarity of academic research careers, OECD Publishing, Paris, #113, doi, (accessed on 18 October 2021). 

R1.3. As the data suggests, there is still a need for both education and culture change in faculties for academic advisors to understand the idea that career paths are very different from their graduate days, that the concept of leaving academia forever is no longer as binary and that CPD in career and transferable is essential for both academic and industry careers It would have been great to touch on this more about how this could also be addressed by universities (and is it being done already?) so that the use of your tool could be better implemented by these stakeholders. Overall, a lovely paper, thank you for your submission.

RA1.3. Thank you for bringing up this important point that an increasing number of universities are embracing. We agree that, if after using the tool to identify weaknesses it is determined that a focus on certain stakeholders such as faculty or university administrators is warranted, we hope it might stimulate conversations among CPD offices and department chairs and deans to better inform faculty of the advantage of transferrable skills for all careers. To address these points in the manuscript we have improved the description of the tool use in the Methods and Results as described below for Reviewers 2 and 3 and added the following specific to this point to the Discussion: 

“Should the stakeholder engagement tool indicate that increased interactions with faculty or university administrators is warranted, that might stimulate conversations among CPD offices, department chairs and deans, to better inform faculty of the advantage of transferrable skills for all careers. These actions will help promote culture change in academia and reinforce awareness of the range of career options beyond academia available to PhDs.” 

Reviewer #2:

R2. PONE-D-21-26337, titled "Using Stakeholder Insights to Enhance Engagement in PhD Professional Development" by Ramadoss et al., outlines the benefits of identifying and engaging different groups of internal and external stakeholders in developing, institutionalizing and communicating value of career and professional development (CPD) for STEM pre-doc and postdoc scholars. In this article, authors develop and utilize a stakeholder engagement visualization tool to inform strategy, priorities and approaches to action for CPD practitioners. As highlighted by the authors, identifying different stakeholders, understanding incentives of engagement and priorities of each is important especially for practitioners in small offices with fewer resources. Also, the attention given to perspectives of stakeholders from diverse demographic backgrounds, who may belong to underrepresented and historically excluded groups is noteworthy. The article is valuable for CPD practitioners in a field that’s growing fast and would benefit from consistent approaches and best practices. I recommend approval with following suggestions and considerations for revisions and improvement.

RA2. We thank the reviewer for these insightful comments and welcome these suggestions for improvement to add value and make the article clearer. We have addressed the need for clarification in the manuscript as noted below each comment.

Major comments

R2.1. Narrative: The authors primary audience are CPD practitioners and their internal stakeholders. According to me, the narrative needs to balance two primary goals: 1. communicating importance of stakeholder engagement to inform and direct CPD efforts, and 2. benefits of utilizing the stakeholder engagement visualization tool. The authors need to better outline the second goal. The result section communicates the importance of stakeholder engagement to inform and direct CPD efforts through text analysis of interview responses. However, the visualization tool itself needs more introduction and context within the text (either in methods or results). Figure 6 was confusing for me. I wish the instructions within the figure 6 were embedded in the text of the article for the reader. The output of the visual seems to corelate to number of stakeholders in each groups and level of engagement. However, I was confused by mention of topics and quadrants. When I hovered on parts of the visual, I could see “A” or “S” but didn’t know what topics “A or S” indicate.

RA2.1. Thank you for helping prompt us to better outline the benefits of using the stakeholder engagement visualization tool and to clarify the interpretation method. As a result, we have made significant additions to both the Methods and Results sections to better clarify the steps to use the stakeholder engagement tool and its interpretation respectively. We could not replicate what the reviewer saw when hovering on the visual, but we have replaced the interactive Excel stakeholder tool file which should hopefully fix that situation. Also, a note has been added to download the file in Excel in order to view and use the tool properly. 

“Note: Please download the Excel file to view proper format.”

The following has been added to the Methods:

“Use of the stakeholder engagement tool

STEP 1: With your unit or collaborators, create a list of stakeholders--start with the suggested categories (see Table 2 below) and customize for your institution & geographic area by entering your own labels to tailor to your institution. Try to be specific (i.e. don't say 'alumni office' but rather, 'alumni association of Boston'). You can further break these down into lists of names and contact info. Note that engaging with naysayers/agnostics can be as valuable as engaging with your supporters to identify blind spots and valuable alternative perspectives.

STEP 2: Based on your institutional knowledge, take one or two minutes with your team to quickly rank your perceived level of engagement with the stakeholder groups in the first column. The authors encourage users to define their own Likert scale to calibrate the level of engagement or use the suggested Likert scale in Table 3.

Table 2: Suggested stakeholders

Internal stakeholders/users faculty supporters, faculty agnostics/nay-sayers, admin staff, graduate 

students, postdocs

Internal partners career services, industry relations, licensing office, communications 

office, alumni office

External partners economic development, chamber of commerce, trade organizations, 

professional societies, funders

Employers foundations/non-profits, local companies, accelerators/incubators, 

intellectual property firms, biopharma

Table 3. Suggested Likert scale

0 No interaction, little/no idea whom to contact

1 Have a short list of contacts, need to reach out, little/no personal connection

2 Reached out to one or more stakeholders, a few productive discussions

3 Know of and work with groups/people to engage, occasional activities together

4 Our offices/institutions/companies engage on a semi-regular basis

5 Well established, regular two-way engagement

For analysis and interpretation of the stakeholder engagement tool, see the Results section.”

Additionally, the following has been added to the Results: 

“To interpret the output of the stakeholder engagement tool, use the Excel file (S5 File; Note: Please download the Excel file to view proper format) which automatically sums each quadrant. The scores toward the left or right of the plot indicate areas of focus externally (right, blue and grey) or internally (left, orange and red). The top half of the plot reveals information on internal/external users (in orange and blue), the lower half on internal/external partners (in red and grey). The sums in each quadrant indicate the relative strength in each stakeholder group (top left: internal stakeholders such as graduate students or faculty; top right: external stakeholders such as employers; bottom left: internal partners such as licensing office; and bottom right: external partners such as professional societies). 

This is a self-reflection tool to identify areas of individual network engagement and areas for potential development. The more the sectors are filled to the outside of the circle, the more perceived relative engagement there is. Based on how you define your Likert scale (see suggested scale in Methods), the score for the relative engagement may reflect representation of stakeholders belonging to each group, as well as frequency and quality of interactions between stakeholders. For example, the CPD office may only have one or two stakeholders in a given stakeholder group, but you might meet with them often and they might be extremely influential and enthusiastic about supporting CPD programs for graduate students and postdoctoral researchers. Moreover, it is important to engage with and address the concerns of naysayers, as this will add value to overall CPD operations and help to launch more successful programming. 

In the example (Fig 6), the internal stakeholders are visually a strength, especially graduate students, and there appears to be room for increased engagement with certain external stakeholders such as intellectual property firms.”

Figure 6 legend has been adjusted to read:

“Fig 6: Stakeholder engagement tool. A rapid assessment tool for internal and external stakeholders to evaluate competencies and determine strengths for engagement in career and professional development programming. Detailed instructions for entering values are in Methods, and interpretation can be found in Results. [see S5 File to download and use the stakeholder engagement tool and view proper format: https://osf.io/fc27x/?view_only=b166987514234b718d8457778651534f]”

R2.2. Perspectives of institutional leaders as key internal stakeholders: Authors mention graduate deans and provosts are potential stakeholders for consideration. The study would benefit from including graduate deans’ perspective. This is especially important as equitable access and institutionalizing CPD comes up multiple times as a theme. While faculty chairs perspective is important, graduate deans have the agency to develop structures and policies, as well as allocate resources to make CPD more accessible and embedded within graduate programs. Their perspective of CPD is a key piece of this puzzle. Similarly, perspectives of representatives from Vice President for Research and/ or Provost division is important for institutionalizing CPD for postdoctoral training for equitable access beyond training grant recipients.

RA2.2. Thank you for this excellent point. While our sample did include graduate deans, we did not include university-wide leaders such as Provosts, Presidents, or Chancellors. We fully agree that a larger and more robust sample of institutional leadership would likely yield unique themes, and we agree that this is a limitation of our sample. Hence in the newly added “Limitations and Future Directions” section of the discussion, we have added the following:

“Faculty and administrators were sampled from a wide variety of roles, ranging from rank-and-file faculty; through departmental leadership such as Chairs, Associate Deans, and other leaders; through leaders in graduate and postdoctoral education such as Deans of Students. Nonetheless, we acknowledge the importance of identifying unique themes and perspectives that may arise for Provosts, Presidents, Chancellors, and other high-level university-wide leadership roles which are crucial to setting strategic aims, funding, and program priorities. The current sample did not include a large enough subset of high-level leaders to analyze the data separately, and future directions of research could include examining this group of administrators specifically.”

R2.3. Approaches to action: In the discussion section, authors highlight the benefits of this engagement resource and utilizing and customizing the tool to create approaches to action. They also summarize some shared themes and incentives across stakeholder groups. To strengthen that point, I would recommend creating a figure with graphic representation of shared themes and ranked barriers or challenges.

RA2.3. Thank you for this suggestion. We have added as a supplemental file a summative list of themes that spanned stakeholders and referenced it in the discussion. This document of shared themes was collated based on the number of mentions of each theme across stakeholders. Therefore, we have added the following text to the discussion and added S6 File:

“In summary, common themes across all stakeholders are shown in S6 File. Many emerging common themes centered on a tailored approach to CPD programs. For example, while most stakeholders acknowledged the need for CPD, the desired timing and content of programming aligned with individual specific needs. This further accentuates the need for creating streamlined access, discussed by most stakeholders.”

Regarding the ranked barriers or challenges, our interviewing process did not determine a priority of each stakeholder’s input, but this is an excellent idea for future work to include in interviews as a feature to the questionnaire. We therefore have added the following text to the “limitations and future research” section:

“Another useful future analysis would be to ask stakeholders to rank which challenges represent the biggest barriers, a function that was not queried in this study’s interviews. Such a ranking would assist CPD offices to prioritize resources to overcome challenges.”

Minor comments

R2.4. Specific guidance and considerations for URM perspectives: Authors indicate percent of interviewees who belong to different racial-ethnic demographic and international population. Given the intersectional identities, could authors draw how many of the interview participants would identify as “under-represented” in their field of study or employment and how their perspectives may provide dimensions to strengths and challenges of CPD? I bring this up as authors provide a good list of identities to consider in each stakeholder group. However, how should CPD practitioners utilize those perspectives to examine nuances to balance with broad priorities. E.g. one interview participant highlights importance of CPD in recruitment of talented graduate students. Studies indicate that CPD is particularly important in recruitment, retention, success and wellbeing of early career scientists from under-represented and historically excluded groups. I wonder whether there’s a scope to analyze the data from this lens after aggregation and making specific recommendations and considerations in ‘approaches to action’. Offering specific guidance on approaching overlapping and distinct themes emerging from perspectives of under-represented and historically excluded stakeholder groups will further highlight the importance of this approach and tool for diverse stakeholders in graduate education, beyond CPD practitioners.

RA2.4. We agree that this is an important issue for future directions of research in this area. Sample sizes of subgroups in any particular intersectional category of race/gender ethnicity with a given stakeholder group were too small to allow for additional analyses. We have added the following in the newly added “Limitations and Future Directions” section:

“Further analyses that merit additional attention include examining unique themes arising through intersectional identity groups (e.g., gender, race, ethnicity, international/domestic status, among others). Unfortunately, sample sizes within each stakeholder group were not large enough to examine these important differences; however this is an important topic for additional work to consider.”

R2.5. How are alums positioned within internal and external stakeholder populations? Authors focused on distinct sets of interview questions for internal and external stakeholders. However, alumni are an interesting population and may provide key perspectives for both sets of questions. Authors could recommend a customized hybrid interview questionnaire for alumni which draws from both internal and external stakeholder interviews. In fact, the alumni survey questions in CGS PhD Career Pathways Project presents a similar hybrid scenario spanning retrospective reflection as a PhD graduate and perspective as a professional.

RA2.5. We agree that alumni perspectives would provide a rich strata of additional data, and while some alumni of author institutions were incidentally included in the sample, doctoral alumni status was not probed separately. This would be an excellent inclusion for future work – and may even merit additional questions beyond either current trainees or the stakeholder group that their later sector of employment falls into – or perhaps a combination of both sets of questions could be illuminating. Accordingly, we have added the following into the newly created “Limitations and Future Directions” section: 

“External stakeholders were sampled from a variety of roles in industry and non-profit employers including small and large companies, intellectual property firms, consultancies, and accelerators. While this study did not specifically sample alumni as a stand-alone stakeholder group, some external stakeholder individuals (e.g., For-Profit, Non-Profit/Society) were incidentally alumni of the institutions represented. Still, we realize it is important to include the voice of recent PhD and postdoctoral alumni for their understanding of how their training affected their employability, how their industry sector views PhDs, and their retrospective views of their careers. This is especially of interest since they are role models for our current students and postdoctoral researchers, and are now potentially in hiring positions. Alumni are a very strong stakeholder group to engage with, as they are eager to give back to their institution, are invested in the institution, and directly experienced programs and the various stakeholders involved in the programs. Hence, investigating this stakeholder group would be extremely useful as a future research topic.”

Reviewer #3:

R3. This paper details a sound methodology for engaging with stakeholders in researcher career development that aims to assist graduate career educators to navigate relevant networks internal and external to their academic institution. By engaging internal-facing networks the methodology allows possible supporters and non-supporters of the career and professional development of researchers within an academic setting to be identified and challenges and opportunities to be identified. The external facing internal stakeholders and external stakeholders all provided feedback on the need for CPD in order to facilitate the future career development of pre- and postdoctoral researchers outside of academia.

RA3. Thank you for this comment, we are glad you found these strengths of the paper to be valuable.

R3.1. In contrast with the very complete description of the stakeholder engagement methodology, the stakeholder visual engagement visualization tool is very scantily described. The authors claim that the tool "is quick and easy to use" without revealing exactly how the visualized data is collected. For example, it is not clear if the scores relate to the number of stakeholder belonging to each group or the quality of their comments or the quality of their support for CPD or of the ease that graduate career educators can access stakeholders from the relevant groups. In Figure 6 "Faculty agnostics/nay-sayers" have a score higher than "faculty supporters", which seems unlikely for quality based responses. Also it appears that "Faculty supporters" is given the color for the wrong quadrant "Employers". Without a description of what the scores represent, it is not clear how the rapid tool can be used to identify challenge areas and strengths. As only stakeholder groups are scored, and not specific challenges and strengths, it is hard to see how this stakeholder visual engagement visualization tool can compete with the time-intensive process described in the rest of the paper that delivers exactly those results. I recommend that the authors resubmit the paper with an improved description of the visualization tool.

RA3.1. We agree that the description of the Stakeholder Tool could be expanded. First, we now better describe usage of the tool and clarify that indeed the tool identifies the level of perceived interactions with stakeholders in different groups and included a suggested Likert scale; further, we have described how strengths and challenges can be identified to expand networks in areas that are found to be proportionally less represented in one’s network. We apologize for the lack of instructions regarding the tool – we could not replicate the color error, but we have replaced the file with the stakeholder tool which should hopefully fix that situation. Also, a note has been added to download and view the file in Excel in order to view the figure properly. As a result of these helpful comments we have accordingly added to the methods, results and discussion clarifying text (as also noted under reviewer #2). In addition, we clarify what the scores relate to with the following text in the Results:

“To interpret the output of the stakeholder engagement tool, use the Excel file (S5 File; Note: Please download the Excel file to view proper format) which automatically sums each quadrant. The scores toward the left or right of the plot indicate areas of focus externally (right, blue and grey) or internally (left, orange and red). The top half of the plot reveals information on internal/external users (in orange and blue), the lower half on internal/external partners (in red and grey). The sums in each quadrant indicate the relative strength in each stakeholder group (top left: internal stakeholders such as graduate students or faculty; top right: external stakeholders such as employers; bottom left: internal partners such as licensing office; and bottom right: external partners such as professional societies). 

This is a self-reflection tool to identify areas of individual network engagement and areas for potential development. The more the sectors are filled to the outside of the circle, the more perceived relative engagement there is. Based on how you define your Likert scale (see suggested scale in Methods), the score for the relative engagement may reflect representation of stakeholders belonging to each group, as well as frequency and quality of interactions between stakeholders. For example, the CPD office may only have one or two stakeholders in a given stakeholder group, but you might meet with them often and they might be extremely influential and enthusiastic about supporting CPD programs for graduate students and postdoctoral researchers. Moreover, it is important to engage with and address the concerns of naysayers, as this will add value to overall CPD operations and help to launch more successful programming. 

In the example (Fig 6), the internal stakeholders are visually a strength, especially graduate students, and there appears to be room for increased engagement with certain external stakeholders such as intellectual property firms.”

We thank our reviewer for indicating that the interviews and stakeholder engagement tool are not mutually exclusive. The tool is not meant to identify challenge areas or strengths within a stakeholder group, but rather serves as a way to focus on strengths of existing relationships, hone in on partners to include in discussions, or optimize outreach to spark new collaborations across stakeholders. 

We would advocate to first use the tool as a preliminary screen to focus on particular stakeholder categories and follow up to develop strong partnerships with stakeholders informed by the themes we have identified through our interviews. To address this very helpful suggestion, we have included the following text in the Discussion:

“The tool serves as a way to focus on strengths of existing relationships, hone in on partners to include in discussions, or optimize outreach to spark new collaborations across stakeholders, rather than to identify specific challenge areas or strengths within a stakeholder group. We would advocate to first use the tool as a preliminary screen to identify which stakeholder categories to focus on. Informed by these themes identified through our interviews, CPD offices can then follow up with stakeholders to develop strong partnerships.”

We thank you for your consideration of our revised manuscript, and look forward to hearing from you regarding your decision.

Sincerely yours,

Deepti Ramadoss, PhD

Assistant Director for Training, Assessment, and Career Exploration

Email: deepti.ramadoss@pitt.edu ● Phone: (412) 383-5246

---

## [Decision Letter · Decision Letter 1]

20 Dec 2021

Using Stakeholder Insights to Enhance Engagement in PhD Professional Development

PONE-D-21-26337R1

Dear Dr. Ramadoss,

We’re pleased to inform you that your manuscript has been judged scientifically suitable for publication and will be formally accepted for publication once it meets all outstanding technical requirements.

Kind regards,

Sina Safayi, D.V.M., Ph.D.

Academic Editor

PLOS ONE

Additional Editor Comments (optional):

Reviewers' comments:

Reviewer's Responses to Questions

**Comments to the Author**

1. If the authors have adequately addressed your comments raised in a previous round of review and you feel that this manuscript is now acceptable for publication, you may indicate that here to bypass the “Comments to the Author” section, enter your conflict of interest statement in the “Confidential to Editor” section, and submit your "Accept" recommendation.

Reviewer #2: All comments have been addressed

Reviewer #3: All comments have been addressed

2. Is the manuscript technically sound, and do the data support the conclusions?

Reviewer #2: (No Response)

Reviewer #3: (No Response)

3. Has the statistical analysis been performed appropriately and rigorously? 

Reviewer #2: (No Response)

Reviewer #3: (No Response)

4. Have the authors made all data underlying the findings in their manuscript fully available?

Reviewer #2: (No Response)

Reviewer #3: (No Response)

5. Is the manuscript presented in an intelligible fashion and written in standard English?

Reviewer #2: (No Response)

Reviewer #3: (No Response)

6. Review Comments to the Author

Reviewer #2: (No Response)

Reviewer #3: (No Response)

7. PLOS authors have the option to publish the peer review history of their article (what does this mean?). If published, this will include your full peer review and any attached files.

Reviewer #2: No

Reviewer #3: No

---

## [Editor Report · Acceptance letter]

19 Jan 2022

PONE-D-21-26337R1 

Using stakeholder insights to enhance engagement in PhD professional development 

Dear Dr. Ramadoss:

I'm pleased to inform you that your manuscript has been deemed suitable for publication in PLOS ONE. Congratulations! Your manuscript is now with our production department. 

Kind regards, 

on behalf of

Dr. Sina Safayi 

Academic Editor

PLOS ONE